# Handling Data Heterogeneity via Architectural Design for Federated Visual Recognition

**Sara Pieri**\* **Jose Renato Restom**\* **Samuel Horvath** **Hisham Cholakkal**

Mohamed Bin Zayed University of Artificial Intelligence (MBZUAI)

## Abstract

Federated Learning (FL) is a promising research paradigm that enables the collaborative training of machine learning models among various parties without the need for sensitive information exchange. Nonetheless, retaining data in individual clients introduces fundamental challenges to achieving performance on par with centrally trained models. Our study provides an extensive review of federated learning applied to visual recognition. It underscores the critical role of thoughtful architectural design choices in achieving optimal performance, a factor often neglected in the FL literature. Many existing FL solutions are tested on shallow or simple networks, which may not accurately reflect real-world applications. This practice restricts the transferability of research findings to large-scale visual recognition models. Through an in-depth analysis of diverse cutting-edge architectures such as convolutional neural networks, transformers, and MLP-mixers, we experimentally demonstrate that architectural choices can substantially enhance FL systems' performance, particularly when handling heterogeneous data. We study 19 visual recognition models from five different architectural families on four challenging FL datasets. We also re-investigate the inferior performance of convolution-based architectures in the FL setting and analyze the influence of normalization layers on the FL performance. Our findings emphasize the importance of architectural design for computer vision tasks in practical scenarios, effectively narrowing the performance gap between federated and centralized learning. Our source code is available at `https://github.com/sarapieri/fed_het.git`.

## 1 Introduction

The growing focus on data privacy and protection [37, 61] has sparked significant research interest in federated learning (FL) as it provides an opportunity for collaborative machine learning in many domains, such as healthcare [16, 14, 42, 55], mobile devices [47, 19], internet of things (IoTs) [53, 25] and autonomous driving [40, 56]. In FL, data is kept separate by individual clients to learn a global model on the server side in a decentralized way.

Despite the potential benefits of federated learning, the FL environments are highly non-trivial, as the local datasets of the individual clients may not accurately represent the global data distribution. As each client is a unique end user, the heterogeneity caused by uneven distributions of features, labels, or an unequal amount of data across clients poses a serious obstacle [27]. Therefore, the setting is inherently challenging since FL methods must accommodate both statistical diverseness and exploit similarities in client data. Typically, the non-independent and identically distributed (non-IID) nature of the clients drastically affects the model's performance, resulting in lower accuracy and slower convergence than the counterparts trained with all the data collected in a single location.

---

\*Equal contribution

37th Conference on Neural Information Processing Systems (NeurIPS 2023).

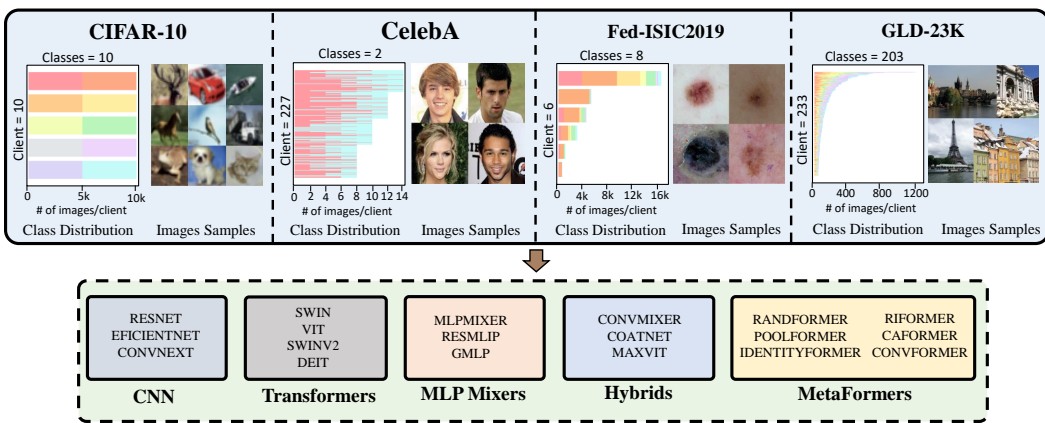

Figure 1: Our study incorporates a wide range of datasets, each presenting varying data heterogeneity levels, including Fed-ISIC2019 and GLD-23K, which are inherently federated datasets due to their image acquisition and sensor mechanisms. Differently from prior literature, we extensively evaluate models from each visual architectural family. Additionally, we assess the effectiveness of four distinct optimization methods orthogonally to the architectural choice. Our findings empirically underscore the significant influence of architectural design choices in narrowing the disparity with centralized configurations and effectively handling data heterogeneity.

To this end, the problem of client heterogeneity received significant attention from the optimization community [23, 38, 58, 29, 36, 1, 75, 50, 15].

In parallel to these developments in FL, visual recognition tasks have witnessed remarkable progress in recent years, primarily due to the advancements in deep learning models. These models have achieved state-of-the-art performance when trained on centralized datasets. However, when deployed in an FL setting, these image recognition architectures often exhibit performance degradation. Recent works have investigated the utilization of pretrained models to address the non-iid issue [4, 54]. Qu et al. [57] examined the robustness of four neural architectures across heterogeneous data, introducing the idea of tackling the problem from an architectural standpoint and encouraging the development of "federated-friendly" architectures. The study suggests that self-attention-based architectures are robust to distribution shifts compared to convolutional architectures, making them more adept in FL tasks. In this work, we perform a comprehensive study across 19 visual recognition models from five different architectural families on four challenging FL datasets. We also re-investigate the inferior performance of convolution-based architectures and analyze the influence of normalization layers on the model performance in non-IID settings.

**Contributions:** This work strives to offer architectural design insights that enable immediate performance improvements without the need for additional data, complex training strategies, or extensive hyperparameter tuning. The key contributions of this work are the following:

- We perform an exhaustive experimental analysis comparing 19 different state-of-the-art (SOTA) models from the five leading computer vision architectural families, including Convolutional Neural Networks (CNNs), Transformers, MLP-Mixers, Hybrids, and Metaformer-like architectures. Some of the architectural families we studied were never previously introduced in FL, demonstrating the feasibility of addressing data heterogeneity in FL from an architectural perspective. These architectures were tested across four prevalent CV federated datasets under the most common non-identicalness sources, such as feature distribution skew, label distribution skew, and domain shift in highly heterogeneous data. Our study, comprehensively illustrated in Figure 1, embodies a step toward more efficient, practical, and robust FL systems.

- Our experiments in highly heterogeneous settings reveal that convolution operations are not inherently sub-optimal in FL applications and Metaformer-like architectures generally exhibit greater robustness in non-IID settings.

- We illustrate that Batch Normalization (BN) adversely impacts performance in the heterogeneous federated learning framework. To counteract this, we empirically demonstrate that

replacing BN with Layer Normalization is an effective solution to mitigate the performance drop. This study case underlines how architectural design choices can significantly influence FL performance.

- We conduct a study on complex networks with four different optimizers, establishing that the application of optimization methods does not yield substantial performance improvements in the context of complex architectures. We argue that altering the architecture, in practical scenarios, offers a more effective and simpler-to-implement choice.

## 2  Related Works

Federated learning (FL) aims to train models in a decentralized fashion, harnessing edge-device computational capabilities to safeguard the privacy of client data. In this setup, clients retain their data on the device, coordinating with a server to develop a global model. A distinct challenge in FL is data heterogeneity, as client data is often non-IID. The issue of addressing this heterogeneity has garnered significant focus from the optimization domain. To enhance the established aggregation method, FedAvg [49], various strategies have emerged. These encompass optimization improvements for federated averaging such as FedAVGM [23], FedProx [38], SCAFFOLD [29], and FedExP [26] and methods to manage data heterogeneity and imbalance like FedShuffle [21], FedLC [34], and MooN [36].

A complementary angle to explore is how the choice of models can elevate FL training outcomes. It has been observed that pretrained models can significantly counteract non-IID challenges [4], [54]. Some previous studies have highlighted the impact of Batch Normalization (BN) on performance drops under heterogeneous settings [39], and few works [22], [24] suggested replacing BN with Group Normalization. Our work is related to Qu et al.'s examination [57] of neural architecture in visual recognition robustness across heterogeneous data splits. Our experiments compare 19 cutting-edge models from diverse computer vision architectural families against demanding heterogeneous datasets and different optimizers.

## 3  Optimization Methods for Federated Learning

In this section, we present a brief overview of the optimization techniques utilized in our study to evaluate their effectiveness in handling data heterogeneity when applied in conjunction with complex computer vision architectures.

**Federated Averaging (FedAVG) [49]:** FedAVG was proposed by McMahan *et al.* (2017) as the standard aggregation method for FL. It offers a straightforward and effective approach where local models are trained on individual devices, and their parameters are averaged to update the global model. However, FedAVG faces challenges in scenarios involving non-IID and unbalanced data, which are prevalent in FL settings.

**FedAVG with Momentum (FedAVGM) [23]** To address some of the limitations of FedAVG, the addition of a momentum term has been explored. This technique enhances the standard FedAVG method by updating the global model at the server using a gradient-based server optimizer that operates on the mean of the client model updates. The inclusion of momentum facilitates efficient navigation through the complex optimization landscape and accelerates learning, particularly in the presence of sparse or noisy gradients.

**FedProx [38]:** To handle the heterogeneity in data and device availability in FL, Li *et al.* proposed FedProx (Federated Proximal). This method introduces a proximal term into the local loss function, preventing local models from deviating significantly from the global model. The proximal term acts as a regularizer, penalizing large deviations from the current global model. This regularization approach improves overall performance and enhances robustness in FL scenarios.

**SCAFFOLD [29]:** SCAFFOLD is a stochastic algorithm designed to enhance the performance of Federated Averaging (FedAVG) by introducing a controlled stochastic averaging step. This step reduces the variance of gradient estimates by correcting the local updates. SCAFFOLD employs control variates (variance reduction) to address the "client-drift" issue in local updates. Notably, SCAFFOLD requires fewer communication rounds, making it less susceptible to data heterogeneity

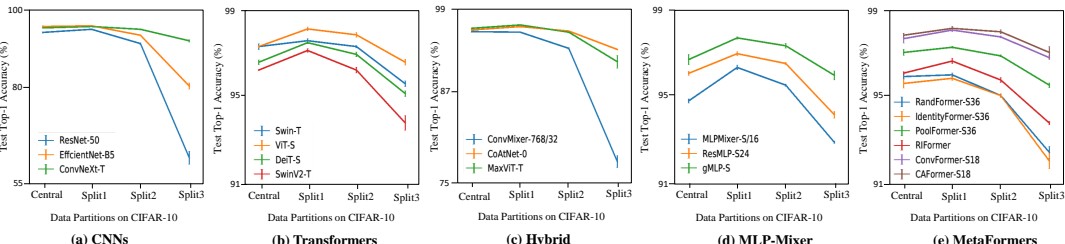

| (a) CNNs | (b) Transformers | (c) Hybrid | (d) MLP-Mixer | (e) MetaFormers |

Figure 2: Model Accuracy Analysis across CIFAR-10 splits grouped by the architecture type. The presented results depict the accuracy performance of various models on distinct partitions of the CIFAR-10 dataset. Each split represents a unique partition, progressing from an IID scenario (Split-1) to a non-IID scenario (Split-3). The models under comparison exhibit similar parameters, FLOPS, and approximately matching ImageNet and centralized accuracies. Notably, Metaformer-like architectures, including Metaformers (e), Transformers (b), and MLP-Mixers (d), exhibit a comparatively smaller performance degradation on Split-2 and Split-3. The best-performing models are CA-Former and ConvFormer, which are Metaformers with advanced token mixers.

or client sampling. Furthermore, the algorithm leverages data similarity among clients, leading to faster convergence.

# 4 Architectures for Computer Vision

In this section, we provide an overview of the architecture families studied in this work.

**Convolutional-based models:** We include Resnet [19] and EfficientNet [62] as the study of Qu *et al.* [57], and extend it by adding a recently introduced architecture, ConvNeXt [45]. EfficientNet and Resnet consist of a stack of convolutional layers, pooling layers, and batch normalization layers. ConvNext was conceived by modernizing ResNet-50 to pair with the performance of vision transformers. The key changes introduced at the component-level are: replacing the typical batch normalization with layer normalization, using inverted residual blocks and larger kernel sizes, and substituting the more common ReLU [51] with the GeLU [20] nonlinearity.

**Vision Transformers:** ViTs were introduced to the computer vision community by Dosovitskiy *et al.* [13] after becoming the de-facto standard for natural language processing tasks. The Transformer architecture reconsiders the need for convolutions and replaces them with a self-attention mechanism. ViTs work by splitting the image into non-overlapping patches treated as tokens and re-weighting them based on a similarity measure between each token pair. The main differences with CNNs are the lack of inductive bias of the vanilla self-attention and the global receptive field capable of capturing long-distance dependencies as opposed to the local receptive field of convolutions. As representatives of this family of architectures, we include ViT [60], Swin [44], Swin V2 [43], and DeiT [66].

**MLP-Mixers:** MLP-Mixers were first proposed by Tolstikhin *et al.* [64]. It was the first architecture to show that although convolutions and self-attention are sufficient to achieve top performance, they are not necessary. The MLP-Mixers take inspiration from ViT's use of patches for input representation but fully use multi-layer perceptrons instead of the typical self-attention mechanism of Transformers. For our experiments, we make use of the original MLP-Mixer [64], the ResMLP variation [65], and the gMLP [41] which uses a special gating unit to enable cross-token interactions.

**Hybrids:** Many efforts have been made to incorporate features from CNNs into Transformers and vice versa. We group this set of mixed models under the same 'Hybrids' category. Namely ConvMixer [67], CoATNet [8], and MaxViT [69]. ConvMixer combines a transformer-like patch design with convolution-based layers to mix patch embeddings' spatial and channel locations. CoATNet vertically stacks convolution layers and attention layers. Finally, MaxViT hierarchically stacks repeated blocks composed of multi-axis self-attention and convolutions.

**MetaFormers:** Although the success of transformer-based architectures was longly attributed to self-attention, recent works [73, 74] have started to identify the overall structure of the transformer block as the critical component in achieving competitive performances. Based on this hypothesis,

Metaformer is constructed from Transformers without specifying the design of the token mixers. If the token mixer is self-attention or spatial MLP, the architecture becomes a Transformer or an MLP-Mixer.

Although most of the Transformers and MLP-Mixers formally fall into the 'Metaformers' structure, we group in this family only the works published by the authors under this name. Within the networks selected, we have some working with simple operators in the token mixers, published with the explicit intent of pushing the limit of the Metaformer structure. Respectively, RandFormer, IdentityFormer [74], and PoolFormer [73] have a random matrix, an identity mapping, and a pooling operator as the token mixer. Moreover, ConvFormer [74] applies depthwise separable convolutions as the token mixer, and CAFormer [74] further combines it with vanilla self-attention in the deep stages. We also add to this category RIFormer [70], an architecture that optimizes the latency/accuracy trade-off by using re-parametrization [12] to remove the token mixer without significantly compromising the performance.

## 5   Experimental Setup

**Models:** This study aims to enrich the FL landscape by introducing a variety of cutting-edge architectures for computer vision. Within the FL framework, we conduct a comparative analysis of models that have similar parameters and FLOPS and roughly equivalent accuracy in centralized settings to the benchmark model, ResNet-50. Ensuring fairness, all the models under consideration fall within a similar parameter range (21-31M) and floating point operations per second (FLOPS) range (4-6G) [1]. Each model exhibits a comparable centralized test accuracy on ImageNet-1K [11].

Unless explicitly stated, all models are pretrained on ImageNet-1K and subsequently finetuned on the specified dataset.

**Federated learning:** In an effort to ensure that the insights derived from this study remain independent of the training heuristics, we have standardized and simplified the training strategy across all architectures. Our approach involves performing FedAVG for a specific number of communication rounds to reach convergence: 100 rounds for both the CIFAR-10 and Fed-ISIC2019 datasets, 30 rounds for the CelebA dataset, and 200 rounds for the GLD-23K dataset. With the exception of the GLD-23K dataset, where we conduct five local steps, each round of local training encompasses one local epoch. The training regimen employs the SGD optimizer with an initial learning rate of 0.03, cosine decay, and a warm-up phase of 100 steps. We set the local training batch size at 32 and apply gradient clipping with a unitary norm to stabilize the training process.

The numerical values presented in this paper represent the average results derived from three repetitions of the same experiment, each with different initializations, to ensure the robustness and reproducibility of our findings. Regarding optimization, we conduct hyperparameter tuning for each model based on the set of values suggested by the respective authors in their original publications. The results reported in this paper reflect the optimal outcomes derived from the hyperparameter search. A more in-depth discussion is given in the supplementary material.

**Datasets:** We conduct experiments on four different datasets, CIFAR-10 [31], CelebA [46], Fed-ISIC2019 [68, 5, 7] and Google Landmarks Dataset v2 (GLD-23K) [72]. Across the experiments, all the images are resized to a standard dimension of 224 × 224. The data heterogeneity of the different datasets used in our experiments is characterized in terms of Label Distribution Skew (LDS), Data Distribution Skew (DDS), and Feature Distribution Skew (FDS). The LDS is characterized by a heterogeneous number of samples per class for each client. In the DDS, the amount of data per client varies. Lastly, the FDS is defined by varying image features per class across clients due to changes in acquisition sensors, image domains, user preferences, geographical locations, etc. (see supplementary materials for more details).

The **CIFAR-10** dataset comprises 60,000 32x32 color images in 10 distinct classes representing common objects. Since CIFAR-10 is not inherently a federated dataset, we artificially simulate three federated data partitions across clients following the approach outlined in [57]. The original test set is retained as the global test set, and 5,000 images are set aside from the training set for validation

---

[1]The only exception in terms of model size is MLPMixer-S/16. However, we have opted to include this model in its smallest configuration provided by the authors, as it constitutes the inaugural exemplar of MLP-Mixer architectures.

|  | CIFAR-10 | | | FED-ISIC2019 | | |
|---|---|---|---|---|---|---|
| ARCHITECTURE | RESNET-50 | CONVNEXT-T | CAFORMER-S18 | RESNET-50 | CONVNEXT-T | CAFORMER-S18 |
| FEDAVG [49] | 59.9 ± 1.87 | 93.5 ± 0.36 | 97.1 ± 0.35 | 67.6 ± 1.59 | 78.6 ± 0.29 | 83.1 ± 0.49 |
| FEDAVGM [58] | 80.2 ± 0.49 | 93.4 ± 0.48 | 97.0 ± 0.14 | 66.2 ± 2.16 | 79.0 ± 0.52 | 83.8 ± 0.37 |
| FEDPROX [38] | 79.0 ± 0.41 | 93.4 ± 0.16 | 97.1 ± 0.25 | 68.0 ± 1.93 | 79.4 ± 1.13 | 83.0 ± 0.79 |
| SCAFFOLD [29] | 78.1 ± 0.06 | 94.3 ± 0.30 | 96.9 ± 0.21 | 62.2 ± 0.26 | 76.0 ± 1.24 | 77.1 ± 0.61 |

Table 1: Accuracy of ResNet-50, ConvNext-T, and CAFormer on CIFAR-10 and Fed-ISIC2019 across four different optimization methods. The average test accuracy of the model is displayed along with the standard deviation of three repetitions of the experiments.

purposes, resulting in a revised training dataset of 45,000 images. We employ the Kolmogorov-Smirnov statistic (KS) to simulate one Independent and Identically Distributed (IID) data partition, namely Split-1 (KS=0), ensuring balanced labels per client, and two non-IID partitions, Split-2 (KS=0.5) and Split-3 (KS=1), with label distribution skew. In Split-2, each client has access to four classes and does not receive samples from the remaining six classes. In Split-3, each client strictly sees samples from two classes only. Therefore, this is the most challenging partition, featuring a pronounced class imbalance. Each data partition has five clients.

The **CelebA** dataset is a large-scale face attributes dataset featuring over 200,000 high-resolution celebrity images. Each image is annotated with 40 binary labels that denote the presence or absence of distinct facial attributes. In this study, we employ the federated version of the CelebA dataset proposed by the LEAF benchmark [3]. In line with the methodology outlined in [57], we test the models on a binary classification task, determining whether a celebrity is smiling. The dataset is partitioned across 227 clients, each representing a specific celebrity and collectively accounting for 1,213 images. The dataset presents significant challenges due to the large-scale setting involving numerous clients. Each client has access to only a handful of samples, five per client on average, and in some instances, data from only one class is available.

The **Fed-ISIC2019** dataset contains skin lesion images collected from four hospitals. We adopt the partitioning strategy proposed by [63], which is based on the imaging acquisition systems utilized. Given that one hospital employed three distinct imaging technologies over time, the federated dataset is distributed across six clients, encompassing 23,247 images. The task involves classifying the images into eight melanoma classes with high label imbalance across clients. Additionally, the dataset exhibits quantity skew with disparity in the number of images per client. The largest client possesses more than half the data, whereas the smallest has only 439 samples.

The **Google Landmark-V2** dataset is a vast collection with over 5 million images of global landmarks. We utilize the federated GLD-23K partition, involving 233 smartphone users as clients contributing 23,080 images across 203 classes. These datasets exhibit a geographically influenced, long-tailed distribution, with some landmarks having many representations while others have very few, mirroring real-world disparities. Also, photographers' images present unique distributional characteristics tied to their habits and locations, resulting in large diversity and imbalance across clients.

## 6 Experimental Results

### 6.1 Optimizers and Complex Networks in Federated Learning

An effective learning system typically requires both an optimized architecture and an effective optimization technique to shape the learning process and resultant model. The federated learning community has focused considerably on optimization techniques to address challenges related to non-IID data and the need to balance statistical diversity and leverage similarities in client data for central model training. However, most recent studies [48] rarely present results incorporating the latest architectures developed by the computer vision community. Many of these publications on vision tasks present their work on simple and shallow networks ranging from a few fully connected layers to the ResNet-50 model. This discrepancy highlights a significant chasm between the research focus of federated learning and state-of-the-art computer vision architectures.

To evaluate the compatibility of advanced architectures with federated optimization methods, we consider the widely used ResNet-50 model in our study and extend it to more recent computer vision architectures. We compare different models with the baseline ResNet-50, combined with diverse optimization techniques across four federated datasets. Alongside ResNet, we employ ConvNext, proposed to narrow the gap of convolutional with transformer architectures, and CAFormer, a recently introduced model in the Metaformer family. We consider four of the most popular optimizers in federated learning, namely FedAVG [49], FedAVGM (enhanced with momentum) [58], FedProx [38], and SCAFFOLD [29].

Results reported in Table 1 can be summarized as follows. First, using the standard FedAVG replacing ResNet-50 with the above computer vision architectures significantly increases accuracy across all datasets, especially with the CAFormer architecture, suggesting that architectural modification can significantly boost performance. Secondly, combining optimizers with complex models does not enhance results, with most of the performance maintaining a steady status, except for FedAVGM applied with ResNet on the CIFAR-10 dataset. In fact, for the Fed-ISIC2019 dataset, some of the optimizers proved to be detrimental to the performance of the network when compared to the simple FedAVG (e.g., SCAFFOLD for all architectures, and ResNet with FedAVGM and FedProx). The lack of optimization benefits across different datasets and architectures can be attributed to the complexity of modern deep learning architectures, such as Transformer-based models, which are significantly more complex than models typically used in FL research, leading to a more challenging optimization landscape. Applying these optimization techniques to the tough non-convex optimization problems encountered during the training of modern deep neural networks remains an unresolved issue. Most FL methods have demonstrated efficacy for certain types of non-convex problems under specific assumptions that might not be applicable to these complex models in practice [10, 30]. Our experiments reveal that the naïve application of these techniques, combined with advanced models, often leads to negligible improvement.

This leaves us with the unresolved question of how we can optimize federated learning performance for practical applications, bridge the gap between advancements in the computer vision and FL communities, and address the issue of closing the performance gap with centralized settings. Inspired by our findings, we inspect the benefits of architectural design choices. Replacing the architecture is an easy-to-implement strategy, given the wide array of pretrained models available and their proven effectiveness on various vision tasks. Moreover, cutting-edge architectures often include design elements that inherently facilitate learning, such as superior feature extraction, incorporation of prior knowledge (inductive biases), and regularization that could mimic the effects of federated optimization techniques. In the following Section 6.2, we explore the possibility of leveraging the inherent strengths of different computer vision models to enhance performance and naturally bridge the non-IID gap in federated learning.

## 6.2 Tackling the Heterogeneity Challenge with Architectural Design

We comprehensively compare network architectures from diverse network families, such as CNNs, Vision Transformers, Hybrids, MLP-Mixers, and Metaformers, with a particular focus on how each architecture family performs against each other. Tables 2 and 3 show our results on CIFAR, CelebA, Fed-ISIC2019, and GLD-23k datasets.

**Architectural Comparison on CIFAR-10:** On CIFAR-10, all models show performance degradation compared to the ideal centralized training, with most of the architectures displaying a significant reduction in accuracy as we progress to the non-IID splits (see Figure 2). Table 2 shows the performance of each model across all splits and the centralized version. The key observations from this study are summarized below.

*Convolutional Networks:* Convolutional models exhibit the overall highest performance degradation among all families. ResNet [19] and EfficientNet [62] are found to be non-robust in heterogeneous settings, with a loss of 36 and 17.1 points, respectively, observed in Split-3. While ConvNeXt [45] shows improvement over its predecessors, it still falls short of achieving top results when compared to other model families.

*Transformers and Metaformers:* Transformers consistently demonstrate robust performance, particularly in non-IID data partitions, exhibiting only a marginal average degradation of 1.7 points. Notably, ViT [60] achieves remarkable resilience, attaining an accuracy of 96.8% and experiencing a minimal drop of merely 0.8%, even in the most challenging split. These findings corroborate the

| MODEL | CENTRAL | SPLIT-1 | SPLIT-2 | SPLIT-3 |
|---|---|---|---|---|
| RESNET-50 [19] | $95.9 \pm 0.21$ | $96.8 \pm 0.17$ | $92.7 \pm 0.25$ | $59.9 \pm 1.87$ (↓ 36.0) |
| EFFICIENTNET-B5 [62] | $97.6 \pm 0.21$ | $97.8 \pm 0.06$ | $95.1 \pm 0.06$ | $80.5 \pm 0.92$ (↓ 17.1) |
| CONVNEXT-T [45] | $97.2 \pm 0.15$ | $97.6 \pm 0.06$ | $96.6 \pm 0.21$ | $93.5 \pm 0.36$ (↓ 3.7) |
| SWIN-T [44] | $97.6 \pm 0.15$ | $97.9 \pm 0.10$ | $97.6 \pm 0.06$ | $95.7 \pm 0.15$ (↓ 1.9) |
| VIT-S [60] | $97.6 \pm 0.06$ | $98.5 \pm 0.10$ | $98.2 \pm 0.12$ | $96.8 \pm 0.15$ (↓ 0.8) |
| SWINV2-T [43] | $96.4 \pm 0.00$ | $97.4 \pm 0.10$ | $96.4 \pm 0.15$ | $93.7 \pm 0.38$ (↓ 2.7) |
| DEIT-S [66] | $96.8 \pm 0.12$ | $97.8 \pm 0.06$ | $97.2 \pm 0.12$ | $95.2 \pm 0.15$ (↓ 1.6) |
| CONVFORMER-S18 [74] | $97.9 \pm 0.12$ | $98.4 \pm 0.12$ | $98.0 \pm 0.12$ | $96.8 \pm 0.35$ (↓ 1.2) |
| CAFORMER-S18 [74] | $98.1 \pm 0.18$ | $98.5 \pm 0.06$ | $\mathbf{98.3 \pm 0.06}$ | $\mathbf{97.1 \pm 0.14}$ (↓ 1.0) |
| RANDFORMER-S36 [74] | $95.7 \pm 0.12$ | $95.8 \pm 0.10$ | $94.6 \pm 0.06$ | $91.3 \pm 0.35$ (↓ 4.4) |
| IDENTITYFORMER-S36 [74] | $95.3 \pm 0.26$ | $95.6 \pm 0.12$ | $94.6 \pm 0.12$ | $90.8 \pm 0.45$ (↓ 4.5) |
| POOLFORMER-S36 [73] | $97.1 \pm 0.25$ | $97.4 \pm 0.10$ | $96.9 \pm 0.06$ | $95.2 \pm 0.12$ (↓ 1.9) |
| RIFORMER-S36 [70] | $95.9 \pm 0.08$ | $96.6 \pm 0.16$ | $95.5 \pm 0.15$ | $93.0 \pm 0.10$ (↓ 2.9) |
| MLPMIXER-S/16 [64] | $94.8 \pm 0.10$ | $96.5 \pm 0.12$ | $95.6 \pm 0.06$ | $92.7 \pm 0.06$ (↓ 2.1) |
| RESMLP-S24 [65] | $96.2 \pm 0.10$ | $97.2 \pm 0.10$ | $96.7 \pm 0.06$ | $94.1 \pm 0.17$ (↓ 2.1) |
| GMLP-S [41] | $96.9 \pm 0.26$ | $98.0 \pm 0.06$ | $97.6 \pm 0.12$ | $96.1 \pm 0.23$ (↓ 0.8) |
| CONVMIXER-768/32 [67] | $97.4 \pm 0.12$ | $97.3 \pm 0.06$ | $94.4 \pm 0.15$ | $74.1 \pm 1.07$ (↓ 23.3) |
| COATNET-0 [8] | $97.7 \pm 0.15$ | $98.3 \pm 0.15$ | $97.5 \pm 0.06$ | $94.2 \pm 0.06$ (↓ 3.4) |
| MAXVIT-T [69] | $98.0 \pm 0.10$ | $\mathbf{98.6 \pm 0.10}$ | $97.3 \pm 0.12$ | $92.0 \pm 1.15$ (↓ 6.0) |

Table 2: Performance of different types of models across all splits of CIFAR-10. The average test accuracy of the model is displayed along with the standard deviation of the experiments. Split-3 shows the degradation compared to the centralized version of the training.

earlier observations made by Qu *et al.* [57] regarding the robustness of ViT models against data heterogeneity. However, Qu *et al.* attribute the superior performance of Transformers, compared to convolutional networks, to the robustness of self-attention in mitigating distribution shifts. In contrast, the emergence of the Metaformer family, employing various token-mixers including convolution operation, suggests that the remarkable proficiency exhibited by Transformers may be primarily attributed to the underlying architectural design of Metaformers, rather than the exclusive reliance on self-attention in models like ViT. Indeed, even Metaformers employing basic token mixers, such as RandFormer with a random token mixer, deliver remarkably satisfactory performances. On the challenging Split-3, CAFormer [74] garners the highest result, achieving an accuracy of 97.1% and delivering exemplary performance across all splits, followed by fully convolutional token mixer (without self-attention) based ConvFormer [74], which is also on par with ViTs.

*MLP-Mixers:* As for MLP-Mixers, on average, their performance on the Split-3 is close to those of Transformers. The best is the gMLP [41] with 0.8% of degradation (second only to ViT) and an accuracy of 96.1%. Finally, Hybrid models achieve reasonable performance, but their results are far behind those of the other families. It is worth noticing that most models achieve higher accuracy in Split-1 compared to the centralized setting. We conjuncture that this is due to a regularizing effect caused by each client's access to the IID data partition (with uniform distribution over 10 classes).

**Architectural Comparison on CelebA, Fed-ISIC2019 and GLD-23k:** The results for CelebA, Fed-ISIC2019 and GLD-23k are reported in Table 3.

*CelebA:* The highest accuracies for CelebA are the ones of Swin [44] (89.0) and ConvFormer [74] (88.1), followed by MLP-Mixer [64] and CAFormer [74]. Again, all Metaformer-like architectures (encompassing Transformers and MLP-Mixers) achieve the best results, far surpassing those of CNNs and Hybrid models, whose average performance is 84 and 74.9, respectively.

*GLD-23K:* On the GLD-23K, we also observe that ViT remains among the highest-performing networks. However, Metaformers such as CaFormer and ConvFormer do not yield high performance as in previous datasets.

*Fed-ISIC-2019:* Finally, on the Fed-ISIC-2019 dataset, which is naturally federated (due to the differences in the image acquisition systems across clients), we find CAFormer and ViT as the best-

| Model | Norm | Main Operation | CelebA | GLD-23K | Fed-ISIC2019 |
|---|---|---|---|---|---|
| ResNet-50 [19] | BN | Conv | 84.9 ± 1.48 | 54.3 ± 0.48 | 67.6 ± 1.59 |
| EfficientNet-B5 [62] | BN | Conv | 79.5 ± 0.50 | 51.9 ± 1.21 | 70.0 ± 0.64 |
| ConvNeXt-T [45] | LN | Conv | 87.5 ± 0.87 | 65.5 ± 0.57 | 78.6 ± 0.29 |
| Swin-T [44] | LN | SA | **89.0 ± 0.79** | 72.1 ± 0.67 | 81.9 ± 0.42 |
| ViT-S [60] | LN | SA | 87.3 ± 0.53 | **76.6 ± 0.69** | 82.3 ± 0.95 |
| SwinV2-T [43] | LN | SA | 86.8 ± 0.62 | 74.4 ± 0.02 | 81.7 ± 0.50 |
| DeiT-S [66] | LN | SA | 87.4 ± 0.60 | 69.1 ± 1.24 | 82.3 ± 0.59 |
| ConvFormer-S18 [74] | LN | Conv | 88.1 ± 0.42 | 53.8 ± 1.11 | 81.1 ± 1.54 |
| CAFormer-S18 [74] | LN | SA+Conv | 87.5 ± 0.49 | 57.8 ± 0.54 | **83.1 ± 0.49** |
| RandFormer-S36 [74] | LN | Conv | 83.9 ± 0.35 | 56.3 ± 0.10 | 77.3 ± 0.23 |
| IdentityFormer-S36 [74] | LN | Conv | 85.8 ± 0.49 | 56.0 ± 2.07 | 76.9 ± 1.43 |
| PoolFormer-S36 [73] | LN | Conv+Pool | 86.4 ± 0.61 | 55.8 ± 1.41 | 79.6 ± 0.29 |
| RIFormer-S36 [70] | LN | Conv | 87.2 ± 0.37 | 69.4 ± 0.20 | 81.9 ± 1.37 |
| MLPMixer-S/16 [64] | LN | Conv | 87.9 ± 0.51 | 71.3 ± 0.32 | 80.5 ± 1.62 |
| ResMLP-S24 [65] | - | Conv | 87.0 ± 0.35 | 64.8 ± 0.59 | 81.1 ± 0.45 |
| gMLP-S [41] | LN | Conv | 86.6 ± 0.31 | 67.4 ± 0.16 | 79.9 ± 1.60 |
| ConvMixer-768/32[67] | BN | Conv | 56.5 ± 1.07 | 45.0 ± 0.76 | 58.6 ± 2.02 |
| CoAtNet-0 [8] | BN+LN | SA+Conv | 82.2 ± 1.66 | 70.4 ± 0.25 | 66.0 ± 3.40 |
| MaxViT-T [69] | LN | SA | 86.1 ± 0.72 | 71.7 ± 0.90 | 70.1 ± 1.27 |

Table 3: Accuracy of the model families on CelebA, GLD-23K, and Fed-ISIC2019. For each model, we report the type of normalization layer, either LayerNorm or BatchNorm (LN/BN), and main operation employed (Convolution/Self-Attention). The mean accuracy of all repetitions is displayed along with the standard deviation.

performing models (83.1 and 82.3 accuracy). In fact, we can see a significantly superior performance from all models following the Metaformer structure, indistinctly of their use of convolutions and/or self-attention in their basic block structure.

In summary, our comprehensive experiments on 19 state-of-the-art architectures demonstrate that the choice of architecture and its components play a noteworthy role in tackling performance degradation caused by data heterogeneity. As a matter of fact, the experiments indicate that selecting an appropriate architecture design can have a greater impact on closing the gap with centralized training than the choice of the optimizer. For example, Table 1 shows that CAFormer with simple FedAVG outperforms all other models even when using more advanced optimization methods.

### 6.2.1 How architectural design improves performances - Normalization Layer

Based on the results observed in the previous sections, we further investigate the performance reduction by delving into the contribution of specific architectural components. Our objective is to identify some key structural elements to achieve stronger performances in heterogeneous settings.

As shown in Table 3, models making use of Batch Normalization (like ResNet, EfficientNet, and ConvMixer) are consistently outperformed by models with Layer Norm on Split-3. This insight sheds light on why older convolutional (such as ResNet and EfficientNet) models have such high degradation. On the contrary, ConvNext, which uses Layer Norm, reaches satisfactory performance, contradicting the initial conclusion of [57] that convolutions are inherently problematic in the federated learning scenario under non-IID conditions. Moreover, we observe how ConvFormer, a model without Self-attention based blocks (based primarily on convolutions) but with a MetaFormer-like structure and Layer Norms, regularly reaches top performance even under the most challenging data partitions.

To further support our claims, we perform experiments with three different models, namely Pool-Former, CoAtNet, and ResNet. Each model trains with a variant using Batch Norm and another using Layer Norm. Lastly, we include a training scheme with FedBN [39] (a method developed specifically to deal with the performance degradation caused by the Batch Norm layer). Results (displayed in Table 4) show a trend indicating the superiority of Layer Norm over Batch Norm, even when using a tailored aggregation method for the latter (e.g., FedBN [39]).

| Variant | Central | Split-3 |
|---|---|---|
| CoAtNet-0 BN | $97.5 \pm 0.04$ | $20.9 \pm 0.45$ ($\downarrow 76.6$) |
| CoAtNet-0 LN * | $97.7 \pm 0.15$ | $\mathbf{94.2 \pm 0.06}$ ($\downarrow 3.4$) |
| PoolFormer-S12 BN | $95.3 \pm 0.35$ | $51.5 \pm 0.32$ ($\downarrow 43.7$) |
| PoolFormer-S12 LN | $95.6 \pm 0.09$ | $\mathbf{91.5 \pm 0.13}$ ($\downarrow 4.1$) |
| ResNet-50 BN | $95.9 \pm 0.21$ | $59.9 \pm 1.87$ ($\downarrow 36.0$) |
| ResNet-50 LN $^{\dagger}$ | $92.8 \pm 0.16$ | $\mathbf{72.2 \pm 3.06}$ ($\downarrow 26.6$) |
| ResNet-50 FedBN [39] | $96.1 \pm 0.19$ | $59.6 \pm 1.40$ ($\downarrow 36.5$) |

Table 4: Accuracy of ResNet-50, CoAtNext-0, and PoolFormer-S12 with BN and LN in the central and the Split-3 version of CIFAR-10. * The original CoAtNet-0 uses BN and LN in convolutional and self-attention blocks, respectively. $^{\dagger}$ LN layers have no pretraining.

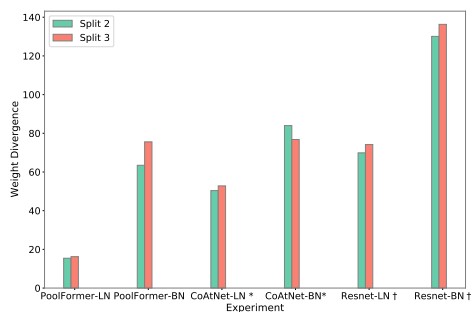

Figure 3: Weight divergence with respect to models' centralized version on CIFAR-10. The use of BN leads to a larger weight divergence.

Finally, as in [76], we measure and report the weight divergence of the models with respect to the centralized training. Figure 3 shows the increase of divergence for the models using BN to their LN counterparts, supporting the selection of Layer Norm over Batch Norm as an architectural choice for federated learning.

# 7 Limitations

Our research primarily explores visual recognition tasks within the federated learning domain, and we have not explored potential downstream tasks, such as object detection and segmentation. It's worth noting that visual recognition models often serve as backbones for extracting input image features in downstream tasks, thereby suggesting that our observed performance trends could likely extend to these downstream tasks. We plan to address this research gap in future studies.

Furthermore, our approach did not design a completely new model from scratch. Instead, we demonstrated our results utilizing pre-existing models and components. Our study also underscores the limitations of employing established optimizers (e.g., FedAVG, FedAVGM, FedProx, SCAFFOLD) along with modern network architectures. However, we abstain from plunging into the creation of innovative optimization techniques. Instead, we focus on evaluating diverse architectural families on many practical federated learning datasets and optimizers. As a result, our investigation leaves an intriguing avenue for future research, integrating our findings to develop dedicated architecture and optimizers tailored specifically for federated learning (FL) scenarios. By exploring this direction, we anticipate the potential to further enhance the performance and efficiency of FL systems.

# 8 Conclusion

This research expands the architectural domain in federated learning, emphasizing architectural design for computer vision applications under data heterogeneity. Our experiments assess various state-of-the-art models from five architectural families across four federated datasets, demonstrating the potential of addressing FL data heterogeneity architecturally. We show that Metaformer-like architectures exhibit superior robustness in non-IID settings. In contrast to existing studies, we observe that convolution operations are not necessarily a sub-optimal choice for architectures in FL applications and highlight the negative impact of Batch Normalization layer on the model performance. Our study also shows that architectural alterations offer an effective and more practical solution to handle data heterogeneity in non-IID settings.

# Acknowledgements

This work is partially supported by the MBZUAI-WIS Joint Program for AI Research (Project grant number- WIS P008).

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

# Supplementary Material

## 9 A Re-look into existing Federated Computer Vision Approaches

In this section, we explore the landscape of computer vision techniques in existing literature. We begin by examining foundational works from the literature and the computer vision architectures they adopt. Our comparative assessment reveals that our approach incorporates several state-of-the-art vision architectures, previously unexplored in federated learning contexts. Furthermore, we contrast commonly used datasets with those we have chosen for our study. Notably, while our methodology encompasses a range of architectures, we also feature five datasets, four of which are intrinsically federated.

| METHOD | VENUE | VISION ARCHITECTURES | VISION DATASETS |
|---|---|---|---|
| FEDAVG [49] | AISTATS'17 | 2MLP, 2CNN | MNIST, CIFAR-10 |
| FEDAVGM [58] | ARXIV'19 | 2CNN | CIFAR-10 |
| FEDPROX [38] | MLSYS'20 | LOGISTIC REGRESSION | MNIST, FEDMNIST |
| SCAFFOLD [29] | ICML'20 | 2MLP | MNIST |
| FEDNOVA [71] | NEURIPS'20 | LOGISTIC REGRESSION, VGG | SYNTHETIC, CIFAR-10 |
| MOON [36] | CVPR'21 | 6CNN, RESNET | CIFAR-10, CIFAR-100, TINY-IMAGENET |
| FEDDYN [1] | ICLR'21 | 6CNN | MNIST, EMNIST, CIFAR-10, CIFAR-100 |
| FEDOPT [58] | ICLR'21 | RESNET, 4CNN, 8MLP | CIFAR-10, CIFAR-100, EMNIST |
| FEDBN [39] | ICLR'21 | 6CNN | SVHN, USPS HULL, SYNTHETIC DIGITS, MNIST |
| MIME [28] | NEURIPS '21 | LOGISTIC REGRESSION, 4CNN, 2MLP | EMNIST |
| FEDLC [34] | ICML'22 | RESNET | SVHN, CIFAR10 AND CIFAR100, IMAGENET SUBSET |
| PROXSKIP [50] | ICML'22 | LOGISTIC REGRESSION | LIBSVM-'W8A' |
| FEDDC [15] | CVPR'22 | LOGISTIC REGRESSION, 2MLP 2CNN, RESNET | MNIST, FASHION MNIST, CIFAR10, CIFAR100 EMNIST, TINY IMAGENET, SYNTHETIC |
| FEDSHUFFLE [21] | TMLR '22 | RESNET | CIFAR-100 |
| FEDEXP [26] | ICLR '23 | 4CNN, RESNET | EMNIST, CIFAR-10, CIFAR-100, CINIC-10 |
| FEDML [17] | NEURIPS WORKSHOP '20 | LOGISTIC REGRESSION, 4CNN RESNET, MOBILENET | MNIST, FEDMNIST, SYNTHETIC CIFAR-10, CIFAR-100, CINIC-10 |
| FLOWER [2] | ARXIV'20 | 4CNN, RESNET | FEMNIST, CIFAR-10, IMAGENET |
| FEDCV [18] | CVPR'21 | EFFICIENTNET, MOBILENET, VIT | CIFAR-100, GLD-23K |
| QU ET AL. [57] | CVPR'22 | RESNET, EFFICIENTNET, VIT, SWIN | RETINA, CIFAR-10, CELEBA |
| SKEWSCOUT [22] | ICML'20 | ALEXNET, GOOGLENET, LENET, RESNET | CIFAR-10, FLICKR-MAMMAL, IMAGENET |
| HSU ET AL. [24] | ECCV'20 | MOBILENETV2, 6CNN | CIFAR-10, CIFAR-100, INATURALIST, GLD-160K |
| **OURS** | | RESNET-50, EFFICIENTNET, CONVNEXT SWIN, VIT, SWINV2, DEIT, RIFORMER-S36, CONVFORMER, CAFORMER, RANDFORMER, IDENTITYFORMER, POOLFORMER MLPMIXER, RESMLP, GMLP, CONVMIXER, COATNET, MAXVIT | CIFAR-10, CELEBA, FED-ISIC2019, GLD-23K, PACS |

Table 5: Literature review and comparative analysis of various federated learning methods evaluated on computer vision benchmarks. Each method is classified by the publication venue, the vision architecture utilized, and the vision-related datasets employed for testing. Furthermore, we provide specific details for shallow models on the complexity of the network architectures in terms of the number of convolutional (CNN) and fully connected (MLP) layers.

| DATASET | IMAGES | CLASSES | CLIENTS | NATURALLY FEDERATED |
|---|---|---|---|---|
| CIFAR-10 [31] | 60,000 | 10 | - | NO |
| CELEBA [46] | 200,288 | 2 | 9,343 | YES |
| FED-ISIC2019 [68, 5, 7] | 23,247 | 8 | 6 | YES |
| GLD-23K [72] | 23,080 | 203 | 233 | YES |
| PACS [35] | 9,991 | 7 | 4 | YES |
| TINYIMAGENET [32] | 120,000 | 203 | - | NO |
| MNIST [33] | 70,000 | 10 | 250 | YES |
| EMNIST [6] | 814,255 | 62 | 500 | YES |
| FEMNIST [3] | 805,263 | 62 | 3500 | YES |
| SVHN [52] | 600,000 | 10 | - | NO |
| CINIC [9] | 270,000 | 10 | - | NO |
| SYNTHETIC DIGITS [59] | 12,000 | 10 | - | NO |

Table 6: Overview of frequently utilized datasets for visual recognition within the field of federated learning. The number of clients indicated for non-naturally federated datasets are subject to each author's unique artificial partition.

### 9.0.1 Data Heterogeneity - Dataset Analysis

In our experiments, we examined data heterogeneity across several datasets, focusing on Label Distribution Skew (LDS), Feature Distribution Skew (FDS), and Data Distribution Skew (DDS). The LDS denotes an uneven distribution of samples across classes for each client, while DDS reflects variations in the volume of data per client. On the other hand, FDS arises from differences in image features per class across clients, attributable to factors like varied acquisition sensors, image domains, user preferences, and geographical locations.

Specifically, for LDS, we tested CIFAR-10 and created three distinct splits with escalating label skew. This skewness was measured using the KS statistic, enabling a comprehensive exploration of how label imbalances affect our model's efficacy.

For Data Distribution Skew (DDS), our primary dataset was GLDK, which exhibited the most pronounced variations in the amount of data per client. This helped us delve deep into the challenges and implications of data volume discrepancies across clients in a federated setting.

In terms of FDS, we incorporated inherently federated datasets like GLD-23K and Fed-ISIC2019, which exhibit this skew. The GLD-23K dataset spans multiple geographical areas, while Fed-ISIC2019 was gathered using diverse medical equipment, offering a broad spectrum of feature distribution skews. Additionally, we delve deeper into the PACS dataset in Sec. 10.4, which showcases significant domain variations. For this particular study, our federated divisions allocated unique target domains: Photos, Sketches, Cartoons, and Paintings.

| DATASET | | # CLASSES | # CLIENTS | # IMAGES | LDS | FDS | DDS |
|---|---|---|---|---|---|---|---|
| CIFAR-10 | SPLIT-1 | 10 | 5 | 60,000 | ✗ | ✗ | ✗ |
| | SPLIT-2 | 10 | 5 | 60,000 | ✓✓ | ✗ | ✗ |
| | SPLIT-3 | 10 | 5 | 60,000 | ✓✓✓ | ✗ | ✗ |
| CELEBA | | 2 | 227 | 1,213 | ✓✓ | ✓ | ✓ |
| FED-ISIC2019 | | 8 | 6 | 23,247 | ✓✓ | ✓✓ | ✓✓ |
| GOOGLE LANDMARK-V2 | | 203 | 223 | 23,080 | ✓✓ | ✓✓ | ✓✓✓ |
| PACS | | 7 | 4 | 9,991 | ✗ | ✓✓✓ | ✗ |

Table 7: Summary of datasets and setups. We assess the data heterogeneity of the datasets in terms of three key aspects: Label Distribution Skew (LDS), Feature Distribution Skew (FDS), and Data Distribution Skew (DDS). We utilize a grading system to quantify the level of skewness, with options ranging from "none" (✗), "mild" (✓), "moderate" (✓✓), to "severe" (✓✓✓).

## 10 Additional Experiments

In this section, we delve into supplementary experiments. These encompass optimization strategies applied to complex network results 10.1, further analysis on normalization layers 10.2, the inclusion of the PASCs dataset 10.4, and a study of convergence speed 10.5.

### 10.1 Optimization Methods with Complex Architectures

| | CELEBA | | |
|---|---|---|---|
| ARCHITECTURE | RESNET-50 | CONVNEXT-T | CAFORMER-S18 |
| FEDAVG [49] | $84.9 \pm 1.48$ | $87.5 \pm 0.87$ | $88.1 \pm 0.61$ |
| FEDAVGM [58] | $84.1 \pm 1.51$ | $87.4 \pm 1.15$ | $87.4 \pm 1.70$ |
| FEDPROX [38] | $85.0 \pm 1.48$ | $87.5 \pm 0.88$ | $87.5 \pm 0.37$ |
| SCAFFOLD [29] | $84.4 \pm 1.81$ | $86.8 \pm 1.33$ | $89.0 \pm 0.40$ |

Table 8: Accuracy of ResNet-50, ConvNext-T, and CAFormer on CelebA across four different optimization methods. The average test accuracy of the model is displayed along with the standard deviation of the experiments. We can appreciate how changing the optimizer achieves virtually negligible improvements, unlike changing the architecture, which yields a significant boost.

## 10.2 Normalization Layer

In this section, we delve into extended experiments concerning normalization layers. We begin by re-inforcing our earlier findings, showcasing that Layer Normalization outperforms Batch Normalization (BN) when tested on an additional dataset, Fed-ISIC2019, as detailed in Table 9.

| VARIANT | FED-ISIC2019 |
|---|---|
| COATNET-0 BN | $68.5 \pm 0.70$ |
| COATNET-0 LN * | $\mathbf{70.4 \pm 0.25}$ |
| POOLFORMER-S12 BN | $64.2 \pm 0.53$ |
| POOLFORMER-S12 LN | $\mathbf{83.2 \pm 0.63}$ |
| RESNET-50 BN | $54.3 \pm 0.48$ |
| RESNET-50 LN [†] | $62.5 \pm 1.50$ |
| RESNET-50 FEDBN [39] | $\mathbf{65.2 \pm 1.24}$ |

Table 9: Accuracy of ResNet-50, CoAtNext- 0, and PoolFormer-S12 with Batch (BN) and Layer (LN) Normalization across the Fed-ISIC2019 datasets. Replacing BN with LN improves performance. * The original CoAtNet-0 uses BN and LN in convolutional and self-attention blocks, respectively. [†] LN layers have no pretraining.

Exploring alternatives to BN, and extending beyond LN, we incorporated Group Normalization (GN) in our assessment. We replicated our results on CIFAR-10, encompassing CoAtNet, PoolFormer, and ResNet - as in the original manuscript—while augmenting it with GN results (averaging three repetitions per experiment along with standard deviations). In this context, CoatNet and PoolFormer exhibited a more pronounced performance drop under GN compared to the LN configuration. Intriguingly, ResNet yielded a converse pattern, achieving superior outcomes with GroupNorm as opposed to LayerNorm. This trend remained consistent while performing the weight divergence analysis across Split-2 and Split-3. These experiments indicate that LayerNorm is a more favorable choice when paired with advanced models. Indeed, the weight divergence in figure 4 serves as a compelling indicator that the incorporation of LN offers a safeguard against drops in performance in heterogeneous partitions, with LayerNorm having inferior weight divergence wrt BatchNorm in all cases and best results for CoAtNet and PoolFormer.

| Variant | CIFAR-10 Central | CIFAR-10 Split-3 |
|---|---|---|
| PoolFormer-S12 BN | $95.3 \pm 0.35$ | $51.5 \pm 0.32$ (↓ 43.7) |
| PoolFormer-S12 GN [†] | $94.1 \pm 0.25$ | $81.5 \pm 0.29$ (↓ 12.6) |
| PoolFormer-S12 LN | $95.6 \pm 0.09$ | $\mathbf{91.5 \pm 0.13}$ (↓ 4.1) |
| ResNet-50 BN | $95.9 \pm 0.21$ | $59.9 \pm 1.87$ (↓ 36.0) |
| ResNet-50 GN [†] | $95.8 \pm 0.19$ | $\mathbf{89.3 \pm 0.59}$ (↓ 6.5) |
| ResNet-50 LN [†] | $92.8 \pm 0.16$ | $72.2 \pm 3.06$ (↓ 26.6) |
| CoAtNet-0 BN | $97.5 \pm 0.04$ | $20.9 \pm 0.45$ (↓ 76.6) |
| CoAtNet-0 GN [†] | $94.3 \pm 0.39$ | $82.6 \pm 2.79$ (↓ 11.7) |
| CoAtNet-0 LN * | $97.7 \pm 0.15$ | $\mathbf{94.2 \pm 0.06}$ (↓ 3.4) |

Table 10: Normalization layer comparison. Accuracy of the models with batch normalization (BN), layer normalization (LN) and group normalization (GN) in the central and the Split-3 version of CIFAR-10. * The original CoAtNet-0 uses BN and LN in convolutional and self-attention blocks, respectively. [†] layers have no pretraining.

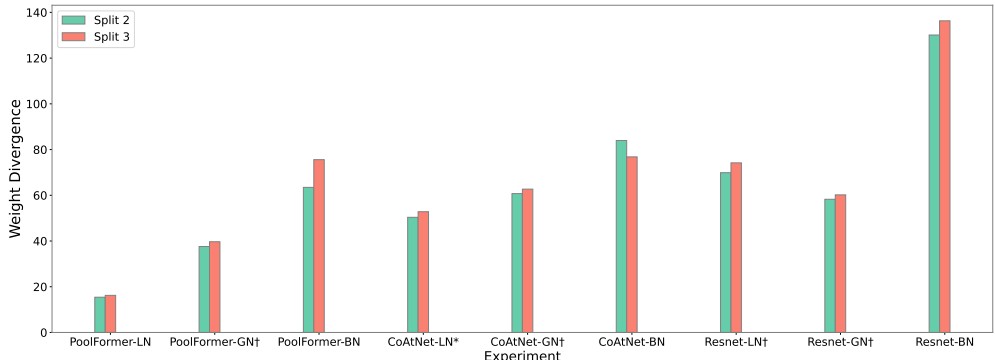

Figure 4: Weight divergence analysis on CIFAR-10. Weight divergence with respect to models' centralized version on CIFAR-10. The use of BN leads to a larger weight divergence. Complex models tend to pair well with LayerNorm, achieving even smaller divergence than when paired with GroupNorm layers.

## 10.3 Centralized Results

| | GLD-23K | | FED-ISIC2019 | |
|---|---|---|---|---|
| MODEL | CENTRAL | SPLIT | CENTRAL | SPLIT |
| RESNET-50 [19] | $79.6 \pm 2.09$ | $54.3 \pm 0.48$ | $88.2 \pm 0.15$ | $67.6 \pm 1.59$ |
| EFFICIENTNET-B5 [62] | $85.3 \pm 0.30$ | $51.9 \pm 1.21$ | $89.7 \pm 0.41$ | $70.0 \pm 0.64$ |
| CONVNEXT-T [45] | $83.7 \pm 0.45$ | $65.5 \pm 0.57$ | $88.1 \pm 0.91$ | $78.6 \pm 0.29$ |
| SWIN-T [44] | $81.9 \pm 0.32$ | $72.1 \pm 0.67$ | $90.8 \pm 0.31$ | $81.9 \pm 0.42$ |
| VIT-S [60] | $79.4 \pm 0.34$ | $\mathbf{76.6 \pm 0.69}$ | $90.3 \pm 0.17$ | $82.3 \pm 0.95$ |
| SWINV2-T [43] | $75.7 \pm 3.92$ | $74.4 \pm 0.02$ | $88.1 \pm 0.58$ | $81.7 \pm 0.50$ |
| DEIT-S [66] | $78.7 \pm 1.04$ | $69.1 \pm 1.24$ | $89.3 \pm 0.36$ | $82.3 \pm 0.59$ |
| CONVFORMER-S18 [74] | $83.7 \pm 0.49$ | $53.8 \pm 1.11$ | $90.3 \pm 0.45$ | $81.1 \pm 1.54$ |
| CAFORMER-S18 [74] | $83.6 \pm 0.31$ | $57.8 \pm 0.54$ | $90.7 \pm 0.22$ | $\mathbf{83.1 \pm 0.49}$ |
| RANDFORMER-S36 [74] | $81.4 \pm 0.27$ | $56.3 \pm 0.10$ | $86.9 \pm 0.23$ | $77.3 \pm 0.23$ |
| IDENTITYFORMER-S36 [74] | $81.3 \pm 0.95$ | $56.0 \pm 2.07$ | $87.0 \pm 0.38$ | $76.9 \pm 1.43$ |
| POOLFORMER-S36 [73] | $83.4 \pm 1.33$ | $55.8 \pm 1.41$ | $88.5 \pm 0.36$ | $79.6 \pm 0.29$ |
| RIFORMER-S36 [70] | $80.7 \pm 1.17$ | $69.4 \pm 0.20$ | $89.5 \pm 0.08$ | $81.9 \pm 1.37$ |
| MLPMIXER-S/16 [64] | $77.5 \pm 0.91$ | $71.3 \pm 0.32$ | $88.6 \pm 0.50$ | $80.5 \pm 1.62$ |
| RESMLP-S24 [65] | $73.4 \pm 1.03$ | $64.8 \pm 0.59$ | $89.1 \pm 0.23$ | $81.1 \pm 0.45$ |
| GMLP-S [41] | $76.5 \pm 0.71$ | $67.4 \pm 0.16$ | $88.6 \pm 0.52$ | $79.9 \pm 1.60$ |
| CONVMIXER-768/32 [67] | $85.6 \pm 0.40$ | $45.0 \pm 0.76$ | $87.3 \pm 0.20$ | $58.6 \pm 2.02$ |
| COATNET-0 [8] | $79.2 \pm 1.03$ | $70.4 \pm 0.25$ | $87.6 \pm 0.17$ | $66.0 \pm 3.40$ |
| MAXVIT-T [69] | $83.4 \pm 0.29$ | $71.7 \pm 0.90$ | $91.0 \pm 0.28$ | $70.1 \pm 1.27$ |

Table 11: Comparative analysis among different architectural families in both federated and centralized settings. Notably, each model experiences a drop in accuracy when transitioning from centralized to federated learning settings. Importantly, high performance in the centralized setting does not necessarily translate to comparable success in the federated split. The table particularly emphasizes the robustness and superior performance of Metaformer-like architectures in the federated setting. These architectures demonstrate a minimal drop in accuracy relative to their centralized counterparts and high overall accuracy results, underlining their adeptness at handling data heterogeneity.

## 10.4 PACS: Analysis on Domain Shift

To simulate a domain shift setting with feature distribution skew, where each client has a target domain with distinct characteristics, we implement a federated version of PACS [35], a 7-class classification dataset commonly used in domain generalization. The total number of images in the dataset is 9991. We partition the data, associating each client with one of the four available domains: Photos, Sketches, Cartoons, and Paintings.

The experiments on PACS [35] are shown in Table 12. On this task, all models reach competitive performances on both the central training and the federated version. The results lead to the aforementioned conclusions, confirming the Metaformer-like models are robust to different types of heterogeneity across the datasets. All models have negligible degradation or show an increase over the centralized version, except for Resnet [19], EfficientNet [62], and ConvMixer [67]. Our hypothesis for such behavior is the combination of the following elements: the small dimension of the target dataset compared to the ImageNet pretraining and the overlapping of classes between both PACS and ImageNet [11].

As we hypothesize that the behavior is caused by the solid pretraining on ImageNet the following question arises: How will the models behave without pretraining? As shown in Table 12, degradation patterns arise again without pretraining. As expected all models have inferior performances, indicating how crucial pretraining is and making it always preferable if weights are available. Moreover, most models with a huge drop in the pretrained version still present the same behavior in the non-pretrained. However, the best-performing models are not those observed in the corresponding pretrained version. The best-performing model is MLP-Mixer [64], along with EfficientNet [62], which has the lowest score with pretraining. As one would expect, the worst-performing models are the ones that require large amounts of data. For example, PoolFormer, IdentityFormer, and RandFormer ([74]) perform very poorly already in the centralized setting, showing that a large dataset is crucial to compensate for the simple token mixers. Likewise, ConvNext [45], and ViT [60] reach a deficient performance on the centralized training version.

| MODEL | CENTRAL-PRETRAINED | SPLIT-PRETRAINED | CENTRAL | SPLIT |
|---|---|---|---|---|
| RESNET-50 | $96.1 \pm 0.26$ | $90.2 \pm 0.53$ ($\downarrow 5.9$) | $82.1 \pm 0.74$ | $42.5 \pm 0.53$ ($\downarrow 39.5$) |
| EFFICIENTNET-B5 | $97.5 \pm 0.05$ | $89.1 \pm 0.45$ ($\downarrow 8.3$) | $87.6 \pm 0.45$ | $60.4 \pm 0.60$ ($\downarrow 27.2$) |
| CONVNEXT-T | $97.2 \pm 0.28$ | $97.6 \pm 0.21$ ($\uparrow 0.4$) | $42.6 \pm 2.30$ | $32.5 \pm 0.34$ ($\downarrow 10.1$) |
| SWIN-T | $97.2 \pm 0.24$ | $97.9 \pm 0.12$ ($\uparrow 0.6$) | $66.8 \pm 0.41$ | $53.6 \pm 0.58$ ($\downarrow 13.2$) |
| VIT-S | $96.9 \pm 0.31$ | $97.9 \pm 0.26$ ($\uparrow 1.0$) | $57.3 \pm 0.96$ | $49.0 \pm 0.53$ ($\downarrow 8.3$) |
| SWINV2-T | $96.4 \pm 0.25$ | $97.7 \pm 0.21$ ($\uparrow 1.2$) | $64.8 \pm 2.62$ | $55.3 \pm 0.95$ ($\downarrow 9.5$) |
| DEIT-S | $96.5 \pm 0.14$ | $97.5 \pm 0.21$ ($\uparrow 1.0$) | $58.0 \pm 2.13$ | $47.7 \pm 0.68$ ($\downarrow 10.4$) |
| CONVFORMER-S18 | $97.8 \pm 0.22$ | $98.2 \pm 0.16$ ($\uparrow 0.4$) | $70.0 \pm 0.24$ | $54.3 \pm 0.63$ ($\downarrow 15.7$) |
| CAFORMER-S18 | $97.9 \pm 0.14$ | $\mathbf{98.5 \pm 0.31}$ ($\uparrow 0.6$) | $68.1 \pm 1.84$ | $50.2 \pm 0.42$ ($\downarrow 17.8$) |
| RANDFORMER-S36 | $95.7 \pm 0.27$ | $95.9 \pm 0.14$ ($\uparrow 0.2$) | $47.9 \pm 1.77$ | $39.6 \pm 0.20$ ($\downarrow 8.3$) |
| IDENTITYFORMER-S36 | $95.4 \pm 0.26$ | $96.1 \pm 0.37$ ($\uparrow 0.7$) | $49.7 \pm 3.41$ | $46.1 \pm 1.37$ ($\downarrow 3.7$) |
| POOLFORMER-S36 | $96.8 \pm 0.20$ | $97.1 \pm 0.40$ ($\uparrow 0.3$) | $46.3 \pm 2.13$ | $37.5 \pm 0.47$ ($\downarrow 8.8$) |
| RIFORMER-S36 | $96.4 \pm 0.14$ | $96.7 \pm 0.26$ ($\uparrow 0.3$) | $55.9 \pm 0.32$ | $40.8 \pm 0.41$ ($\downarrow 15.0$) |
| MLPMIXER-S/16 | $96.0 \pm 0.45$ | $96.5 \pm 0.45$ ($\uparrow 0.4$) | $70.1 \pm 0.14$ | $\mathbf{63.2 \pm 0.59}$ ($\downarrow 6.9$) |
| RESMLP-S24 | $96.2 \pm 0.38$ | $96.4 \pm 0.03$ ($\uparrow 0.3$) | $64.1 \pm 0.72$ | $35.0 \pm 2.56$ ($\downarrow 29.1$) |
| GMLP-S | $96.3 \pm 0.41$ | $97.9 \pm 0.24$ ($\uparrow 1.6$) | $71.6 \pm 0.17$ | $59.4 \pm 0.26$ ($\downarrow 12.2$) |
| CONVMIXER-768/32 | $97.3 \pm 0.36$ | $75.7 \pm 5.31$ ($\downarrow 21.6$) | $69.8 \pm 1.84$ | $20.7 \pm 0.38$ ($\downarrow 49.1$) |
| COATNET-0 | $97.0 \pm 0.29$ | $94.7 \pm 0.29$ ($\downarrow 2.3$) | $79.8 \pm 0.71$ | $47.7 \pm 0.59$ ($\downarrow 32.1$) |
| MAXVIT-T | $97.3 \pm 0.25$ | $96.7 \pm 0.16$ ($\downarrow 0.6$) | $83.4 \pm 0.74$ | $50.9 \pm .041$ ($\downarrow 32.6$) |

Table 12: A comprehensive evaluation of the accuracy achieved by various model families on the PACS dataset under different operational configurations. This includes centrally stored data and federated learning scenarios, with and without ImageNet pretraining. The results offer a detailed insight into the relative effectiveness of the model families under these varying conditions, providing a clear understanding of the impact of pretraining on model performance.

## 10.5 Convergence Speed

The efficiency of convergence is crucial in the context of federated learning. The inherent complexity of this learning paradigm imposes communication overhead for each update iteration, reinforcing the need for rapid convergence. Fast convergence — reaching the optimal model parameters in fewer iterations — minimizes communication demands, reducing bandwidth usage and overall training time. Faster convergence plays a significant role in real-world federated learning applications where devices may be intermittently available or have unreliable connections. We show that architecture selection (architectural choices) has a significant impact on the convergence speed in federated learning scenarios. The rate of convergence among models varies notably. Notably, the best-performing models we highlighted, specifically ViT and MetaFormers, also exhibit accelerated convergence in most of the experiments. Surprisingly, within the GLDk-23 set, CAFormer lags with the slowest convergence rate, suggesting a need for more in-depth exploration of complex large-scale settings in future studies.

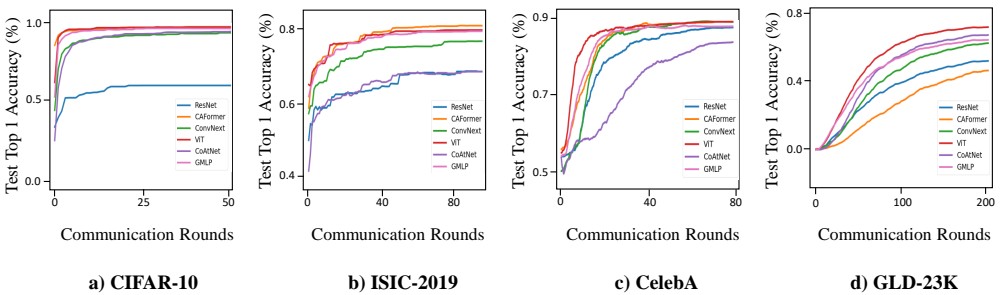

Figure 5: Comparative evaluation of convergence rates for different neural network architectures across diverse datasets. (a) CIFAR-10, (b) Fed-ISIC2019, (c) CelebA, and (d) GLD-23K. Convergence time is measured in terms of communication rounds. This comparison highlights the significant reduction in convergence time enabled by architectural changes.

# 11 Implementation Details

## 11.1 Machines

We simulate the federated learning setup (1 server and N devices) on a single machine with 32 Intel(R) Xeon(R) Silver 4215 CPU and 1 NVidia Quadro RTX 6000 GPU.

## 11.2 Hyperparameter Search

Our study only tests a limited set of hyperparameters for the optimization methods, mostly adhering to the respective original authors' recommendations. For FedProx, we conduct a hyperparameter search across a spectrum of $\mu$ values, specifically $\mu \in \{10^{-4}, 10^{-3}, 10^{-1}, 1\}$ keeping the clients learning rate at $\eta = 3 \times 10^{-2}$.

For SCAFFOLD, we perform a hyperparameter search over a grid of client learning rates $\eta \in \{10^{-4}, 3 \times 10^{-4}, 10^{-3}, 3 \times 10^{-3}, 10^{-2}, 3 \times 10^{-2}, 10^{-1}, 3 \times 10^{-1}, 1\}$ with global learning rate $\eta_{gl} = 1$. Hyperparameter tuning is harder for FedAvgM as it involves an additional hyperparameter for momentum $\beta$. In our search, we considered $\beta \in \{0, 0.5, 0.7, 0.9, 0.99\}$ and server learning $\eta_{sr} \in \{10^{-4}, 3 \times 10^{-4}, 10^{-3}, 3 \times 10^{-3}, 10^{-2}, 3 \times 10^{-2}, 10^{-1}, 3 \times 10^{-1}, 1\}$. The learning rate of the client optimizer is held constant at $\eta = 3 \times 10^{-2}$.

Although additional tuning of the regularization parameters may sometimes yield improved empirical performance, we do not anticipate a significant impact on the results discussed. This demonstrates the advantage of architectural design choices over off-the-shelf optimization methods, which can be integrated with minimal modifications to the training process.

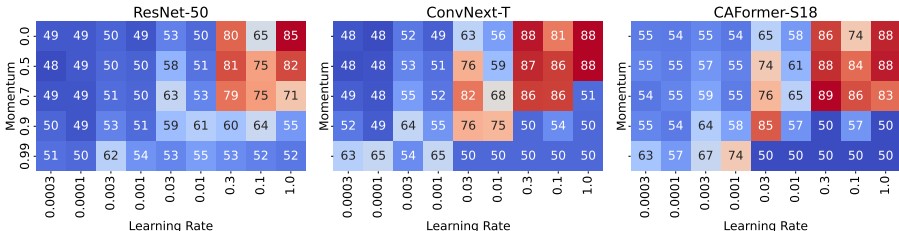

Figure 6: Visualization of the results from a hyperparameter grid search for the FedAvgM algorithm on the federated version of CelebA dataset. Three different architectures were tested: ResNet, ConvNext, and CAFormer. The explored hyperparameters include momentum and server learning rate.

## 11.3 Architectures

Ensuring comparable models is indeed crucial for obtaining fair results. All of the architectures tested (with the exception of MLP-Mixer) have a similar size, measured in number of parameters (21-31M), and similar computational complexity, measured in FLOPS (4-6G). Other elements that might impact the computational cost, input size, etc., are also kept constant.

Regarding the variations in pre-trained accuracy on ImageNet, there are slight differences in performance among the models. However, it is essential to consider that this diversity is inevitable due to the wide range of architectures we incorporated that span across several years of research. To address fairness in our comparisons, we analyzed the absolute numerical result of the experiments and discussed the drop in performance for each model compared to its own centralized version of the training. Furthermore, all models perform similarly in the centralized setting on CIFAR-10, while their performances significantly differ in challenging splits. Both analysis allows us to assess the relative impact of each model's modifications while accounting for the individual model's baseline.

| MODEL | PARAM | FLOPS | IMAGENET-1K ACC |
|---|---|---|---|
| RESNET-50 | 26M | 4.1G | 76.2 |
| EFFICIENTNET-B5* | 30M | 2.4G | 83.6 |
| CONVNEXT-T | 29M | 4.5G | 82.1 |
| SWIN-T | 29M | 4.5G | 81.3 |
| VIT-S* | 22M | 4.5G | 83.1 |
| SWINV2-T* | 28M | 4.6G | 83.5 |
| DEIT-S | 22M | 4.6G | 79.8 |
| CONVFORMER-S18 | 27M | 3.9G | 83.7 |
| CAFORMER-S18 | 26M | 4.1G | 84.1 |
| RANDFORMER-S36 | 31M | 5.2G | 79.5 |
| IDENTITYFORMER-S36 | 31M | 5G | 79.3 |
| POOLFORMER-S36 | 31M | 5G | 81.4 |
| RIFORMER-S36 | 31M | 5G | 81.3 |
| MLPMIXER-S/16* | 59M | 13G | 76.4 |
| RESMLP-S24 | 30M | 6G | 79.4 |
| GMLP-S* | 20M | 6G | 79.6 |
| CONVMIXER-768/32* | 21M | 20G | 80.16 |
| COATNET-0 | 25M | 4.2G | 81.6 |
| MAXVIT-T | 31M | 5.6G | 83.6 |

Table 13: Comparative analysis of examined architectures based on key performance and complexity metrics. The metrics include the number of parameters, Floating Point Operations Per Second (FLOPS), and ImageNet-1K performance results. *FLOPS were computed using the Fvcore library.

