# OpenReview forum: "Handling Data Heterogeneity via Architectural Design for Federated Visual Recognition"
_NeurIPS.cc/2023/Conference — NeurIPS 2023 poster_

### Official Review · Reviewer_6dpE · 2023-06-30

**Soundness:** 3 good
**Presentation:** 3 good
**Contribution:** 2 fair
**Rating:** 5
**Confidence:** 4

**Summary:**

This paper conducted extensive experiments to show the effect of the neural architecture design on non-iid federated learning (FL) for visual recognition. More specifically, experimental results demonstrated that applying the SOTA neural architecture design to FL can significantly mitigate the performance degradation incurred by non-iid data distribution. Moreover, it also shows that layer normalization performs better than batch normalization in non-iid FL.

**Strengths:**

1. The paper is well-motivated. It bridges the gap between the outdated architecture utilized in the FL research community and the SOTA architecture in visual recognition.

2. The authors have some interesting observations, such as the critical roles of layer normalization and the larger effect that the architecture can have over the optimizer.

**Weaknesses:**

1. The paper lacks technical novelty. It just simply applies existing SOTA visual models to non-iid FL. If the authors could utilize their observation to design an architecture with a customized optimizer, which achieves better performance, then the technical novelty will be improved significantly.

2. It would be better if the author could conduct more fine-grained studies, like the effect of architecture on feature skew setting, label skew setting and convergence rate of the models.

**Questions:**

1. Although it seems to be mentioned, I want to confirm whether the evaluated models with different architectures have similar model size, computational costs and pre-trained accuracy on ImageNet.

2. For the experiments in Table 2 and 3, do they use FedAvg for model aggregation?

**Limitations:**

The study of the effect of neural architecture on non-iid FL is limited to visual recognition, which limits its impact.

---

> ### Author Rebuttal · Authors · 2023-08-09
>
> We thank the reviewer for the encouraging and insightful comments. **Please find our responses to specific queries below.**
>
> >**1. On technical novelty**
>
> Rather than introducing an entirely novel approach to federated learning (FL), our paper focuses on conducting thorough experiments using state-of-the-art (SOTA) visual recognition architectures and network components to provide insights and practical guidelines for effectively tackling FL data heterogeneity within the context of modern computer vision architectures. We hope that  our study will serve as a catalyst for the creation of innovative FL strategies tailored for SOTA computer vision architectures.
>
> We would like to clarify that our paper includes not only experiments on various SOTA models but also architectures from different families of backbones for visual tasks, along with a component-level analysis. Additionally, we performed an in-depth analysis of common optimization techniques for non-iid FL on SOTA visual recognition models, which is a contribution in itself. Moreover, we perform experiments on five datasets among which four are naturally federated. As suggested by the reviewer, we intend to use this study as a foundation for developing a custom architecture with an optimized approach in the future.
>
> Please note that we have provided a comprehensive summary of key insights and practical takeaways from our paper in the *common response to all reviewers* provided above.
>
> >**2.  More fine-grained studies, like the effect of architecture on feature skew, label skew, and convergence rate .**
>
> Thank you for your insightful comment. Regarding label distribution skew (LDS), we have conducted experiments on CIFAR-10 and curated three different splits that exhibit increased label skew, quantified using the KS statistic. This approach allows us to thoroughly investigate the impact of label imbalance on our models' performance.
>
> As for feature distribution skew (FDS), we have included challenging naturally federated datasets, particularly GLD-23K and ISIC2019, that present such heterogeneity. These datasets encompass diverse geographical locations (GLD-23K) and utilize various medical equipment for data acquisition (ISIC2019), providing a rich representation of feature distribution skew. Also, we included in Supplementary Material a dedicated analysis of the PACS dataset, which exhibits notable domain shifts. For this investigation, our federated partitioning involved the allocation of distinct target domains (Photos, Sketches, Cartoons, and Paintings).
>
> Concerning the convergence rate of our models, we have dedicated a section (Sec. 2.5) in the Supplementary Material - Additional Experiments to present the result. We found that Mataformer-like architectures, ViT, and CAFormer exhibit faster convergence rates compared to other methods. As suggested, these fine-grained experiments will be included and highlighted in the final draft.
>
> >**3. Although it seems to be mentioned, confirm whether the evaluated models with different architectures have similar model size, computational costs and pretrained accuracy on ImageNet.**
>
> Ensuring comparable models is indeed crucial for obtaining fair results. All of the architectures tested (with the exception of MLP-Mixer) have a similar size, measured in number of parameters (21-31M), and similar computational complexity, measured in FLOPS (4-6G). Refer to Table 7 in Supplementary Material - Implementation Details. Other elements that might impact the computational cost, input size, etc., are also kept constant.
>
> Regarding the variations in pre-trained accuracy on ImageNet, there are slight differences in performance among the models. However, it is essential to consider that this diversity is inevitable due to the wide range of architectures we incorporated that span across several years of research. To address fairness in our comparisons, we analyzed the absolute numerical result of the experiments and discussed the drop in performance for each model compared to its own centralized version of the training.
> Furthermore, all models perform similarly in the centralized setting on CIFAR-10, which makes it interesting to analyze the FL setup as their performances significantly differ in challenging splits.
> Both analysis allows us to assess the relative impact of each model's modifications while accounting for the individual model's baseline.
>
> >**4. For the experiments in Table 2 and 3, do they use FedAvg for model aggregation?**
>
> Yes, Tables 2, 3, and 4 use FedAvg for model aggregation. Experiment results combining optimizers and complex networks are in Table 1. Due to minor gains and inferior performance, subsequent experiments (after Section 5.1) adopted standard FedAvg. This will be better clarified in the final version. Additional results on optimization techniques are in Supplementary Material's Section 2, Table 3.
>
> >**5. The study of the effect of neural architecture on non-iid FL is limited to visual recognition.**
>
> We acknowledge that our research specifically centered on studying the impact of neural architecture in the context of non-IID FL for visual recognition tasks. While visual recognition significantly influences various vision tasks and real-world scenarios, we agree that FL applies to diverse tasks as well. To overcome this limitation and improve our work's influence, we plan to expand our research in forthcoming projects by investigating other downstream tasks like segmentation and object detection. It's worth noting that visual recognition models often serve as backbones for extracting input image features in downstream tasks, thereby suggesting that our observed performance trends could likely extend to other tasks. We plan to incorporate this point into the limitations section of our revised manuscript, as shown in the "Sec. B. Discussion on limitations*  listed in the *common response to all reviewers*.

---

> > ### Comment · Reviewer_6dpE · 2023-08-12
> > **Confirmation of Reading the Rebuttal**
> >
> > Thanks for the well-organized rebuttal from the authors. The feedback solves some of my questions and make the key points of the paper more clear to me. I think the authors made some early and preliminary explorations of the impact of the neural architecture on non-iid FL for visual recognition. But the novelty of the technique and the insights from the experiments are still a little bit limited. Therefore, I tend to keep my current rating.

---

> > > ### Author Response · Authors · 2023-08-14
> > >
> > > Thank you for acknowledging the clarity of our rebuttal and the preliminary exploration highlighted in our paper. We genuinely value your feedback. To provide further clarity on our contributions, we'd like to provide a broader perspective:
> > > 1. In-depth Analysis Over Broad Range: Our study goes beyond merely introducing novel architectures to FL. We dissect the intricate relationships between architectural choices and FL's nuances. While some might perceive the exploration of 19 visual recognition models as incremental, understanding the myriad combinations of datasets, architectures, and optimization methods holistically indeed fills a critical gap in the literature.
> > > 2. Pioneering Guidelines for FL Architectures: Our paper not only presents outcomes but also provides valuable insights into the practical application of our findings, culminating in actionable guidelines for the FL community. This fusion of experimental results with pragmatic advice underscores the broader applicability of our discoveries.
> > > 3. Contrast with Prior Studies: We made pointed efforts to showcase how our findings, particularly concerning normalization layers and convolutions in non-IID settings, build upon and, in some instances, challenge previous understandings. Our research serves as a starting point for future studies, urging them to consider our observations while framing their experiments.
> > > 4. Real-world Implications of Findings: FL is rapidly advancing, and its real-world applications are burgeoning. By delineating the resilience of specific architectures and dispelling myths about others, our research can significantly influence how FL models are built and deployed in practical settings.
> > >
> > >
> > > While we understand the benchmark for novelty can vary, our aim has consistently been to offer a granular, comprehensive, and actionable understanding of FL's architectural intricacies. We believe this depth of exploration, combined with the breadth of our study's scope indeed offers fresh insights and a novel contribution to the FL realm.
> > > We genuinely hope our perspective resonates with yours and helps emphasize the novelty and broader significance of our research.

---

> > > > ### Comment · Reviewer_6dpE · 2023-08-21
> > > > **Final response to the authors**
> > > >
> > > > Thanks to the detailed explanation from the authors. I think I fully understand the key points of the paper and I prefer to keep my original rating.

---

### Official Review · Reviewer_TNv8 · 2023-07-06

**Soundness:** 3 good
**Presentation:** 4 excellent
**Contribution:** 3 good
**Rating:** 6
**Confidence:** 4

**Summary:**

This paper presents an experimental analysis comparing several state-of-the-art neural networks architectures on four different federated vision datasets in highly heterogeneous settings using multiple different optimization approaches.

**Strengths:**

Thorough experimental study: Four datasets, four optimizers, numerous architectures.

Useful experimental study:
This paper conducts experimental study in FL setting with several latest neural network architectures used in the computer vision community. This study takes a step in the direction of bridging the gap between advances in FL and CV communities. This study also provides insights into the roles of architecture and optimizer in FL setting.
1) Advanced optimizers may not be useful when latest/advanced neural network architectures are used.
2) Choosing an appropriate architecture design can have a greater impact on closing the gap with centralized training than the choice of the optimizer.
3) Metaformer-like architectures are more resilient to data heterogeneity. Self-attention is not necessarily the key component.
4) Convolutions may not be inherently problematic in non-iid FL settings. The type of normalization used matters a lot.





**Weaknesses:**

Various existing works have already emphasized the problem BatchNorm causes in FL settings, and proposed to use GroupNorm instead of BatchNorm For example, "The Non-IID Data Quagmire of Decentralized Machine Learning" and "Federated Visual Classification with Real-World Data Distribution". This paper proposes to replace BN it with LayerNorm for CNNs, but has not provided any experimental results comparing it with GroupNorm.


Typo:
Table 1 - Caption mentions CelebA but there are no results on CelebA in the table.

**Questions:**

Is LayerNorm better than GroupNorm in FL settings?

---

> ### Author Rebuttal · Authors · 2023-08-09
>
> We thank the reviewer for the encouraging and insightful comments. **Please find our responses to specific queries below.**
>
> > **1.1 *(Weakness)* Various existing works have already emphasized the problem BatchNorm causes in FL settings, and proposed to use GroupNorm instead of BatchNorm, for example...
> 1.2 *(Question)* Is LayerNorm better than GroupNorm in FL settings?**
>
> The papers mentioned by the reviewer (The non-iid data quagmire of decentralized machine learning - Hsieh et al., 2020; Federated visual classification with real-world data distribution - Hsu et al., 2020) acknowledge Batch Normalization's challenges in federated learning scenarios and substitute BatchNorm with Group Normalization for convolutional neural networks.
>
> In particular, the paper "The Non-IID Data Quagmire of Decentralized Machine Learning"  by Hsieh et al. delves into BatchNorm's limitations and explores Group Normalization as a viable alternative. While the research demonstrates an enhanced result for GroupNorm on LeNet over CIFAR-10, it lacks quantitative analysis for other choices, such as LayerNorm, and results on different architectural paradigms, especially non-convolutional architectures.
>
> Indeed, we did not show results on Group Normalization in our manuscript. As we performed experiments to assess the robustness of different architectures, the preliminary results showed that models with BatchNorm had a higher drop in accuracy compared to centrally stored data and inferior performances compared to models with LayerNorm. In particular, the best results were achieved by Transformer and Metaformer models, as LayerNorm was introduced with the advent of Transformer architectures. Replacing BatchNorm with LayerNorm in methods utilizing convolution operations (eg: CoAtNet, ResNet) allowed us to demonstrate that convolutions are not inherently suboptimal in heterogeneous settings (in contrast with the observations of Qu et al., 2022). For this experiment, Layer norm and Barch norm were the natural choices because those two layers are the most commonly used across SOTA architectures, affording a natural Convolution vs. Self Attention evaluation.  As suggested by the reviewer, we have extended our BatchNorm/LayerNorm experiments in this rebuttal by incorporating the GroupNorm layer.
>
> We replicated our results on CIFAR-10, encompassing  CoAtNet, PoolFormer, and ResNet - as in the original manuscript—while augmenting it with GN results (averaging three repetitions per experiment along with standard deviations). In this context, CoatNet and PoolFormer exhibited a more pronounced performance drop under GN compared to the LN configuration. Intriguingly, ResNet yielded a converse pattern, achieving superior outcomes with GroupNorm as opposed to LayerNorm. This trend remained consistent while performing the weight divergence analysis across Split-2 and Split-3.  These experiments indicate that LayerNorm is a more favorable choice when paired with advanced models. Indeed, the weight divergence serves as a compelling indicator that the incorporation of LN offers a safeguard against drops in performance in heterogeneous partitions, with LayerNorm having inferior weight divergence wrt BatchNorm in all cases and best results for CoAtNet and PoolFormer.
>
> Our findings are presented in the table below, detailing the results, along with an analysis of weight divergence in Image 1 reported in the attached PDF document.
>
> | **Variant**        | **Central** | **Split-3** |
> |:------------------:|:---------------------:|:---------------------:|
> |  PoolFormer-S12 BN |   $95.3 \pm 0.35$    | $51.5 \pm 0.32$ (↓43.7) |
> |  PoolFormer-S12 GN† |   $94.1 \pm 0.25$   | $81.5 \pm 0.29$ (↓12.6) |
> |  PoolFormer-S12 LN |   $95.6 \pm 0.09$   |  $91.5 \pm 0.13$ (↓4.1) |
> |  ResNet-50 BN     |   $95.9 \pm 0.21$    | $59.9 \pm 1.87$ (↓36.0) |
> |  ResNet-50 GN†   |   $95.8 \pm 0.19$   |  $89.3 \pm 0.59$ (↓6.5) |
> |  ResNet-50 LN†    |   $92.8 \pm 0.16$    | $72.2 \pm 3.06$ (↓26.6) |
> |  CoAtNet-0 BN      |   $97.5 \pm 0.04$     | $20.9 \pm 0.45$ (↓76.6) |
> |  CoAtNet-0 GN†    |   $94.3 \pm 0.39$   | $82.6 \pm 2.79$ (↓11.7) |
> |  CoAtNet-0 LN *    |   $97.7 \pm 0.15$    |  $94.2 \pm 0.06$ (↓3.4) |
>
> *Note*:  Accuracy of the models with BatchNorm (BN), LayerNorm (LN), and GroupNorm (GN) in the central and the Split-3 version of  CIFAR-10. * The original CoAtNet-0 uses BN and LN in convolutional and self-attention blocks, respectively. † layers have no pretraining.
>
> > **2. Typo: Table 1 - Caption mentions CelebA but there are no results on CelebA in the table.**
>
> Thank you for bringing the typo to our attention. CelebA results for Table 1 were indeed reported in Supplementary Material Section 2.1. We acknowledge the oversight in Table 1's caption and will remove the excessive text in the final draft.
> The paper's conclusions from Table 1 are in synchrony with the results on CelebA. We observed that altering the optimizer along with complex models led to virtually negligible improvements. Conversely, when we modified the architecture, we achieved a substantial boost in performance. Refer to the main paper, Section 5.1 - Optimizers and Complex Networks in Federated Learning, for an in-depth discussion.

---

> > ### Comment · Reviewer_TNv8 · 2023-08-14
> > **Thank you for the rebuttal and additional experiments.**
> >
> > After reading the rebuttal, I stick to my initial rating which is in favor of accepting the paper.

---

### Official Review · Reviewer_JN2W · 2023-07-06

**Soundness:** 4 excellent
**Presentation:** 4 excellent
**Contribution:** 3 good
**Rating:** 6
**Confidence:** 4

**Summary:**

This paper focuses on exploring the effects of architectural design choices on the performance of federated learning (FL) systems in visual recognition tasks, particularly when faced with data heterogeneity. To accomplish this, the paper conducts a comprehensive experimental analysis involving 19 state-of-the-art models from five architectural families. The experiments are conducted on four federated datasets, and the results are compared against centralized training and various optimization methods. Additionally, the paper investigates the impact of normalization layers on the performance of FL systems. It provides insights into the role of normalization layers and their influence on FL performance, ultimately proposing guidelines for selecting appropriate architectures in FL scenarios.

**Strengths:**

1, This paper offers a comprehensive investigation by considering a diverse set of models, datasets, and optimization methods, enabling a thorough evaluation of their performance in the presence of varying levels of data heterogeneity. By conducting this wide-ranging analysis, the paper provides a fair and comprehensive comparison of these approaches.
2, This paper introduces several novel architectures, including Metaformers, MLP-Mixers, and Hybrids, which have not been extensively explored in the federated learning (FL) literature. These architectures are shown to achieve comparable or superior performance to more traditional models such as Transformers and CNNs in FL settings.
3, The writing is clear and easy to follow.


**Weaknesses:**

1, While this paper provides a comprehensive study of architectural design for FL, no novel methods are developed in this paper.
2, This paper lacks a clear definition or measure of data heterogeneity, which hampers the ability to compare results consistently across different datasets and splits. Without a well-defined metric, it becomes challenging to quantify and understand the extent of data heterogeneity in the experiments conducted.



**Questions:**

1, It could be better to provide a discussion on how the findings are related to other key factors such as communication costs in FL. For example, for two given architectures, does the model with lower performance always have better communication efficiency? Can a single model provide good performance and communication efficiency at the same time?

**Limitations:**

Please refer to the weakness and questions.

---

> ### Author Rebuttal · Authors · 2023-08-09
>
> We thank the reviewer for the encouraging and insightful comments. **Please find our responses to specific queries below.**
>
> >**1. While this paper provides a comprehensive study of architectural design for FL, no novel methods are developed.**
>
> We acknowledge that our work does not propose new FL methods. Instead, it offers a thorough exploration of architectural design in FL. We believe our study to shed light on the various insights on architectural design that can drive future advancements in FL. We plan to leverage the knowledge gained to pursue future work in creating novel FL-specific architectures that can address challenges and applications in the domain. We have provided a comprehensive summary of key contributions in Sec. A of *common response to all reviewers* given above.
>
> >**2.  Lacks a clear definition or measure of data heterogeneity...**
>
> Thank you for the opportunity to clarify this point. The standard approach in the FL community involves evaluating methods on artificially generated datasets, where non-IID splits are created by selecting a heterogenous number of samples per class for each client. The sampling strategy is typically performed either according to a Dirichlet distribution with different $\alpha$ levels, where the parameter $\alpha$ controls how much the partition is unbalanced, or Kolmogorov-Smirnov (KS) statistic to quantify the difference data distributions of data across clients as in [Ensemble distillation for robust model fusion in federated learning - Lin et al., 2020], [No fear of heterogeneity: Classifier calibration for federated learning with non-iid data - Luo et al., 2021] and [Rethinking.. - Qu et al., 2022]. In our experiments, we used this second metric for the CIFAR-10 dataset, for which we reported increased KS statistic value for three splits over which we have direct control of the label allocation.
>
> After assessing our results on simulated non-IID splits with CIFAR-10, according to what was done by current literature, we evaluated our results on four additional naturally federated datasets (see Appendix for PACS dataset), showcasing the consistency of our findings. This second approach embodies a more challenging setting that aligns closely with real-world scenarios, an aspect we strongly advocate for in FL research.
>
> We acknowledge the lack of a clear metric to quantify the data heterogeneity in real-world datasets.
> Indeed for datasets that already present a natural federated partition, there is no standard metric in the community to assess the data heterogeneity. In particular, because the data exhibit different heterogeneities apart from the label distribution skew, those are typically intertwined in real-world scenarios and can be highly non-trivial to measure (e.g., the above-mentioned metrics cannot be straightforwardly applied). We considered, in addition to the label distribution skew, feature distribution skew, where the image features per class vary across clients, and data distribution skew, where the amount of data per client varies. We believe this categorization provides a good estimation of the diverse types of heterogeneity that real-world data may exhibit in the wild.
>
> The below table indicates the data heterogeneity of different datasets used in our experiments in terms of  Label Distribution Skew (LDS), Feature Distribution Skew (FDS), and Data Distribution Skew (DDS). The LDS is characterized by a heterogeneous number of samples per class for each client. In the DDS, the amount of data per client varies. Lastly, the FDS is defined by varying image features per class across clients due to changes in acquisition sensors, image domains, user preferences, geographical locations, etc.  Furthermore, apart from the dataset description in Section 4, we refer readers to Image 1 of the manuscript for a visual representation of the client's data and class distribution. We will incorporate this discussion in the revised manuscript.
>
> |**Dataset**| | **# Classes** | **# Clients** | **# Images** | **LDS** | **FDS** | **DDS** |
> |:--:|:--:|:-:|:-:|:-:|:-:|:-:|:-:|
> |**CIFAR-10**|**Split-1**|10| 5|60,000|✘|✘|✘|
> | |**Split-2**|10|5|60,000|✔✔|✘|✘|
> | |**Split-3**|10 |5|60,000|✔✔✔|✘|✘|
> |**CelebA** | |2|227 |1,213|✔✔|✔|✔|
> |**ISIC2019** | |8|6|23,247|✔✔|✔✔|✔✔|
> |**Google Landmark-V2** | | 203| 223 |23,080|✔✔|✔✔|✔✔✔|
> |**PACS**| | 7|4|9,991|✘|✔✔✔|✘|
>
> *Note*: Summary of datasets and setups. We assess the data heterogeneity of the datasets in terms of three key aspects: Label Distribution Skew (LDS), Feature Distribution Skew (FDS), and Data Distribution Skew (DDS).  We utilize a grading system to quantify the level of skewness, with options ranging from "none" (✘), "mild" (✔), "moderate" (✔✔), to  "severe" (✔✔✔).
>
> > **3.  How the findings are related to other factors, such as communication costs.? ... Can a single model provide good performance and communication efficiency?**
>
> We recognize the importance of communication costs within FL settings. We purposefully chose models with approximately the same dimension (parameters and FLOPS ) and upheld consistency in other aspects that affect communication costs (including communication rounds, protocol, and device types) to ensure a fair comparison. Moreover, our architectural modifications led to a negligible impact on communication costs. By design, this approach helps to maintain fair communication complexity across all evaluated methods. For this reason, we did not explicitly analyze communication complexity.
>
> Nevertheless, as elaborated in the Supplementary Material (Sec 2.5), the rate of convergence among models varies notably. Models with faster convergence necessitate fewer communication rounds to reach saturation. In practical scenarios, this could translate to a reduction in communication rounds, minimizing communication costs. Notably, the best-performing models we highlighted, specifically ViT and MetaFormers, also exhibit accelerated convergence in our experiments.

---

> > ### Comment · Area_Chair_X5eE · 2023-08-18
> >
> > Thank the authors for the rebuttal. PCs and I have reminded the reviewers to respond to the rebuttals as soon as possible. The final decision will depend on both the reviews and rebuttal.
> >
> > @Reviewer JN2W: This message is yet another reminder. Please try to respond to the rebuttal asap.
> >
> > --AC

---

### Official Review · Reviewer_vstz · 2023-07-07

**Soundness:** 3 good
**Presentation:** 2 fair
**Contribution:** 3 good
**Rating:** 7
**Confidence:** 3

**Summary:**

In this comprehensive research study, the authors dive into the realm of federated learning (FL) and its application in visual recognition. They address a crucial aspect often overlooked in FL literature - the impact of architectural design choices on achieving optimal performance. While FL offers collaborative training without exchanging sensitive data, it presents challenges in matching the performance of centrally trained models. The authors conducted an extensive analysis of various cutting-edge architectures including convolutional neural networks, transformers, and MLP-mixers, evaluating their effectiveness in FL systems. By experimenting with 19 visual recognition models from five architectural families across four challenging FL datasets, the authors highlight the significant role of architectural choices in enhancing FL performance, particularly when handling heterogeneous data.

**Strengths:**

For Computer Vision tasks, it is proven that an appropriate architecture design can have a greater impact. However, for Federated Learning, this work fills up such a gap that prior statement can be applicable here too, with centralized training.

**Weaknesses:**

1. Can the figure captions be made small? Instead of writing everything on caption, it is better to write these on the actual paper.
2. Figure 2: It is better to reposition this adjacent to Table 2. And the discussion of splits and central from here should be reflected after Section 4 (after Dataset). And it should be discussed how uniquely it has been partitioned.
3. Table 1, third line: “with the standard deviation of the experiments.” - Here, the total number of experiments should be added, in number.
4. Line 273: “....observations made by Qu et al….”- It is better to add what the observations were made in the paper, at least in short, for better readability.
5. Table 3 caption: LN/BN - The full form can be added prior/adjacent to this, so that it can be better read.
6. Referring to #2 of the Questions section, it needs to be mentioned.

Minor:
1. Naming convention system. If you are naming/specifying/describing something in a specific way, it is better to follow that same thing throughout the paper. For instance, ISIC2019, from line 193, is not written in the same manner in Line 230.


**Questions:**

1. Line 162: Can you please describe why you used this cosine weight decay? It should be added to the paper, and can be described in a line or more.
2. Line 280: “On the challenging Split 3,.....” - You mentioned this Split as challenging, but what special this Split has which makes it challenging?


**Limitations:**

Limitations not added; however, I would suggest adding it.

---

> ### Author Rebuttal · Authors · 2023-08-09
>
> We thank the reviewer for the encouraging and insightful comments. **Please find our responses to specific queries below.**
>
> >**1. Smaller figure captions?....**
>
> We thank the reviewer for the feedback. We will address this in the final draft.
>
> >**2. Reposition  Figure 2 adjacent to Table 2....**
>
> We thank the reviewer for the suggestion. We will reposition Fig. 2 adjacent to Table 2 in the final draft and include the discussions in Section 4.
>
> >**3. Table 1, ... the total number of experiments should be added.**
>
> We have repeated each experiment three times and reported the standard deviation. We will revise the captions in Table 1 and other tables to include this repetition number.
>
> >**4. Line 273: “....what are the observations made by Qu et al….”.**
>
> We appreciate your suggestion. In line 273, we will enhance the passage by summarizing the key observations made by Qu et al. This addition will provide readers with a clear and concise comprehension of the context. Notably, it will emphasize that the Metaformer and ConvNext experiments exhibit robust performance even with convolutional operations.  This stands in contrast to the insights presented by Qu et al. that convolutional operations relying on local high-frequency patterns are potentially problematic in non-IID FL settings, and self-attention-based architectures are more robust in FL settings.
>
> >**5. Table 3 caption: Adding the full forms of LN/BN**
>
> The full form of Batch normalization (BN) and Layer Norm (LN) is currently defined in lines 326-332. This will be defined in Table 3 caption as well for better readability.
>
> >**6. Referring to #2 of the Questions section, it needs to be mentioned.**
>
> In CIFAR-10, the heterogeneity stems from distinct simulated data partitions with different label distribution skew. Split-3, the most demanding, confine each client's labels to only two of the ten classes, introducing a significant label skew (KS=1). We will include this explanation in the manuscript as suggested. For more comprehensive details, please refer to the response provided in *Question 9*.
>
> >**7. Minor: Consistency in naming... For instance, ISIC2019, in line 193 and Line 230.**
>
> We thank the reviewer for pointing out this. The dataset will be addressed as "ISIC2019"  in the final draft. We will also check for other naming convention mismatches.
>
> >**8. Line 162: describe why you used this cosine weight decay?.**
>
> Apologies for the typo. In line 162, we indeed referred to the cosine learning rate decay. The decision to utilize a learning rate decay was based on its demonstrated effectiveness in achieving optimal solutions within the FL domain, as highlighted in several papers, including (On the convergence of fedavg on non-iid data. - Li et al, 2019). Specifically, the selection of the cosine decay is grounded in the fact that cosine learning rate decay is a common choice in the training recipe across most of the families of architectures selected (Are transformers more robust than cnns? - Bai et al., 2021; Bag of tricks for image classification with convolutional neural networks - He et al., 2019; Early convolutions help transformers see better - Xiao et al., 2021). We will promptly update the final draft to provide a concise description of the cosine learning rate decay for further clarity.
>
> >**9. Line 280: “On the challenging Split 3,.....”  what makes this split challenging?**
>
> The reason Split-3 is considered challenging is due to the simulated heterogeneity in the data partition. As CIFAR-10 lacks a natural per-client data partition, we have devised distinct splits to introduce varying degrees of label distribution skew. As mentioned in lines 178-183 of the Experimental Setup - Dataset section, in Split-1, the data is uniformly distributed across client labels. In Split-2, each client has access to only four out of the ten classes, introducing more heterogeneity. Split-3 is the most challenging because each client is limited to samples from only two classes. This restriction creates a significant label skew, making it the most difficult split on the CIFAR-10 dataset. To quantify the heterogeneity, we use the Kolmogorov-Smirnov statistic (KS), and Split-3 has a value of 1, indicating the highest level of label skew compared to other splits. The below table indicates the data heterogeneity of different datasets used in our experiments in terms of  Label Distribution Skew (LDS), Feature Distribution Skew (FDS), and Data Distribution Skew (DDS). The LDS is characterized by a heterogeneous number of samples per class for each client. In the DDS, the amount of data per client varies. Lastly, the FDS is defined by varying image features per class across clients due to changes in acquisition sensors, image domains, user preferences, and geographical locations, etc. We will incorporate this discussion in the revised manuscript to enhance the reader's understanding.
>
> |**Dataset**| | **# Classes** | **# Clients** | **# Images** | **LDS** | **FDS** | **DDS** |
> |:--:|:--:|:-:|:-:|:-:|:-:|:-:|:-:|
> |**CIFAR-10**|**Split-1**|10| 5|60,000|✘|✘|✘|
> | |**Split-2**|10|5|60,000|✔✔|✘|✘|
> | |**Split-3**|10 |5|60,000|✔✔✔|✘|✘|
> |**CelebA** | |2|227 |1,213|✔✔|✔|✔|
> |**ISIC2019** | |8|6|23,247|✔✔|✔✔|✔✔|
> |**Google Landmark-V2** | | 203| 223 |23,080|✔✔|✔✔|✔✔✔|
> |**PACS**| | 7|4|9,991|✘|✔✔✔|✘|
>
> *Note*: Summary of datasets and setups. We assess the data heterogeneity of the datasets in terms of three key aspects: Label Distribution Skew (LDS), Feature Distribution Skew (FDS), and Data Distribution Skew (DDS).  We utilize a grading system to quantify the level of skewness, with options ranging from "none" (✘), "mild" (✔), "moderate" (✔✔), to  "severe" (✔✔✔).
>
> >**10. Limitations not added....**
>
> Please see Sec. B in the *common response to all reviewers* above, with the title "Discussions on Limitations". This discussion on limitations will be incorporated in the final draft of our paper.

---

> > ### Comment · Reviewer_vstz · 2023-08-14
> > **Thanks for the rebuttal**
> >
> > Thanks for the well-organized rebuttal from the authors. The feedback solves my questions and makes the key points of the paper more clear to me. So, I am increasing my rating to "accept" from "weak accept".

---

### Official Review · Reviewer_nuGC · 2023-07-09

**Soundness:** 3 good
**Presentation:** 3 good
**Contribution:** 2 fair
**Rating:** 5
**Confidence:** 4

**Summary:**

This work conducted extensive empirical evaluation to study the influence of model structure on federated learning performance. The paper evaluated the performance of 19 models from 5 model families on 4 different federated learning datasets. The results showed that architecture design has a significant impact on federated learning performance.

**Strengths:**

1.	The work is well motivated. Studying the impact of architecture design on federated learning is important to improve the performance and inspire new federated learning methods.

2.	There are extensive experiments to evaluate the performance of different models.

3.	The paper is easy to follow.


**Weaknesses:**

1.	I appreciate that the author conducted such extensive empirical evaluation. It takes a lot of efforts. However, there are not many insights and takeaways from these evaluations. It seems that the major information this paper tries to convey is that architecture design has critical impact on the federated learning performance. This is not a instructive conclusion and not surprising, as one of the most important thing in deep learning is the model architecture. I suggest the author to conclude more insights and guidelines for future works on designing architectures in federated learning. And it would be better to give some fundamental explanations on why some designs are better rather than just based on empirical evaluations.

2.	There should be some introduction to related works. What studies have been done in previous works? How is this work related to them and different from them? What new insights does this work bring to the community?


**Questions:**

Please see the weakness section.

**Limitations:**

There is no discussion about limitations

---

> ### Author Rebuttal · Authors · 2023-08-09
>
> We thank the reviewer for the encouraging and insightful comments. **Please find our responses to specific queries below.**
>
> > **1. I appreciate that the author conducted such an extensive empirical evaluation. It takes a lot of effort. ,..., I suggest the authors conclude more insights and guidelines for future works on designing architectures in FL...**.
>
> We thank the reviewer for acknowledging the efforts of our extensive experiments. Although it is true that the importance of architecture design is known in the deep learning field, in the federated learning community, a great number of papers still evaluate the results on very shallow or non-state-of-the-art architectures such as ResNet.  Our study embodies a step toward more efficient, practical, and robust FL systems. Furthermore, most approaches to tackle data heterogeneity by the FL community have been from an FL optimization standpoint, and only limited efforts have been put into the architectural design aspect of the problem.  In this work, we demonstrate that architectural design is important in the FL field and necessary to tackle real-world problems with challenging datasets without increased communication costs.  We also perform fine-grained experiments such as weight divergence analysis (Fig 3 in the paper), convergence speed (Sec 2.5 in the Supplementary material), etc., to better explain the improved performance of certain design choices.  We have provided a comprehensive summary of key insights and practical takeaways from our paper in Sec. A of *common response to all reviewers* provided above.
>
> > **2. There should be some introduction to related works. What studies have been done?...**
>
> We would like to thank the reviewer for the suggestion. In the current draft, we have included a brief discussion about related approaches in the introduction (lines 38-45 in the paper). Moreover, a literature review and comparative analysis of nearly 20 federated learning approaches evaluated on computer vision benchmarks is provided in the supplementary material Sec. 1, Tables 1 and 2.  In our Suppl. Table 1, each method is classified by the publication venue, the vision architecture utilized, and the vision-related datasets employed for testing. Furthermore, we provide specific details for shallow models on the complexity of the network architectures in terms of the number of convolutional (CNN) and fully connected (MLP) layers. This comparative evaluation shows that our method covers various state-of-the-art vision architectures that are never studied in the federated learning literature across five datasets (among which four datasets are naturally federated). As suggested, we will include a related work section along with Tab. 1 suppl. in the final draft. The related work section will highlight the contributions of existing works and explain how our work builds upon and differs from them, as briefly outlined below.
> - The growing focus on data privacy and protection (Li et al., 2021b; Shokri & Shmatikov, 2015) has sparked significant research interest in Federated Learning (FL) as it provides an opportunity for collaborative machine learning in various domains. Despite its potential benefits, FL faces fundamental challenges for real-world applications. The FL environments are complex, as local datasets of individual clients may not accurately represent the global data distribution. Heterogeneity caused by uneven feature distributions, labels, or data amounts across clients presents a significant challenge. FL methods need to accommodate statistical diversity and exploit similarities in client data. The non-IID (Independent and Identically Distributed) nature of clients can affect model performance, resulting in lower accuracy and slower convergence compared to centrally-trained models.
> - The problem of client data heterogeneity has received attention from the optimization community (Lian et al., 2017; Li et al., 2019b; 2020; Karimireddy et al., 2020; Acar et al., 2021; Mishchenko et al., 2022; Gao et al., 2022).
> - An orthogonal approach is investigating how architectural choices can enhance FL training performance. Pretrained models have been shown to alleviate non-IID issues (Chen et al., 2022a; Nguyen et al., 2022).
> - Some previous studies have highlighted the impact of Batch Normalization on performance drops under heterogeneous settings (Li et al.; 2021), and few works (Hsieh et al., 2020; and Hsu et al., 2020) suggested replacing BN with Group normalization. However, these studies often used shallow or non-state-of-the-art architectures. We argue that analyzing more complex and modern models in FL settings is essential.
> - Our work is related to Qu et al.'s (2022) examination of neural architecture robustness across heterogeneous data splits. Unlike their work, we show that convolution-based architectures are not inherently inferior to transformer-based ones in non-IID settings. Our experiments compare 19 state-of-the-art models from different computer vision architectural families on challenging heterogeneous datasets and optimizers. Based on this extensive analysis, we propose architectural modifications like replacing batch normalization with layer normalization to improve FL performance significantly.
> - Our study demonstrates that state-of-the-art FL optimization techniques may not provide optimum performance when combined with state-of-the-art complex networks. In practical scenarios, altering the architecture offers a more effective and simpler choice.
>
>  > **3. What new insights does this work bring to the community?**
>
> Kindly refer to our reply to *question 1* and the "Key Contributions and Insights" provided in Sec. A of *common response to all reviewers*.
>
>  > **4. Discussion about limitations**.
> Please see Sec. B  of *common response to all reviewers*  above, under the title "Discussion on Limitations". This discussion on limitations will be incorporated into the final draft of our paper.

---

> > ### Comment · Reviewer_nuGC · 2023-08-12
> > **Thanks for the rebuttal**
> >
> > Thank the author for the detailed rebuttal. My concerns about the insightful takeaways from the paper are partially addressed. I increased the score to borderline accept.

---

### Author Rebuttal · Authors · 2023-08-09

We thank the reviewers for the positive comments. In this rebuttal, we have exerted our best efforts to address nearly all of the reviewers' comments, resulting in numerous enhancements to our final draft. *These improvements include*: (i) additional experiments on using Group normalization, (ii) experiments to evaluate convergence speed, and weight divergence, (iii) dedicated sections to discuss limitations and related works, (iv) additional details on data heterogeneity of different datasets.  *Our code and trained models will be made publicly available.*

##**A. The key contributions and insights of our paper are summarized below:**
1. **Impact of Architecture Design on FL Performance (Acknowledged by Reviewers nuGC, vstz, TNv8, JN2W)**: The paper provides an extensive evaluation of 19 visual recognition models across diverse datasets and optimization methods. It highlights the importance of understanding architecture's impact on federated learning (FL) performance and addressing gaps in FL literature. For example,
given comparable complexity to a ResNet-50, by replacing the architecture, we achieve an increase of more than 37 points in accuracy on the most challenging split-3 of CIFAR-10. Moreover, we perform experiments on 5 challenging datasets, including non-naturally federated datasets such as CIFAR-10, naturally federated datasets such as ISIC2019, Google landmark-V2, CelebA, and Domain generalization dataset PACS (supplementary material).

2. **Introduction of Novel Architectures and Bridging with SOTA Visual Recognition Architecture
(Acknowledged by Reviewers JN2W, 6dpE, TNv8)**: The paper introduces novel architectures such as Metaformers, MLP-Mixers, and Hybrids to the FL literature. Moreover, this work contributes towards bridging the architecture used in FL research with state-of-the-art visual recognition architecture. It also proposes guidelines and suggestions for selecting appropriate architectures in FL scenarios.

3. **Insights on Normalization Layers (Acknowledged by Reviewers 6dpE, JN2W, TNv8)**: Our paper shows that the use of Batch Normalization hinders the performance in FL non-iid setting, to which the paper proposes an effective off-the-shelf solution: replacing it with layer normalization, typically used in the self-attention mechanism. Furthermore, our experiments in this rebuttal comparing layer norm, batch norm,  and group normalization show that layer norm consistently performs well across various architecture families, including state-of-the-art Metaformers.  Our study modernizes the prior research (SkewScout- ICML20, and Hsu et al. - ECCV20) recommending group normalization in FL setting based on their experiments on convolutional neural networks and shallow networks, highlighting the broader applicability of Layer norm in FL non-IID setting.

4. **Effect of Architecture over FL Optimizers on Closing Gap with Centralized Training  (Acknowledged by Reviewers TNv8, 6dpE)**: The paper demonstrates that advanced FL optimizers may not provide optimum performance when using the latest neural network architectures, emphasizing architecture's role in closing the gap with centralized training.

5. **Resilience of Metaformer-like Architectures to Data Heterogeneity  (Acknowledged by Reviewer TNv8)**: The paper highlights the resilience of Metaformer-like architectures to data heterogeneity and the insight that self-attention isn’t the sole key component. Moreover, we analyze how the best-performing models, namely ViT, CAFormer, and ConvFormer, also tend to possess a faster convergence speed (See Supplementary material Sec 2.5 and Fig. 1). Once again, vouching for the use of MetaFormer-like architectures for FL.

6. **Validation of Convolutions in Non-IID FL Settings  (Acknowledged by Reviewer TNv8)**: Our study demonstrates that convolutions may not be inherently problematic in non-iid FL settings. This is in contrast with a previous study (Qu et al. - CVPR 2022) suggesting that convolution operations are sensitive to local high-frequency patterns, potentially leading to inferior performance in heterogeneous FL scenarios. Our findings underline the adaptability of convolutional architectures in diverse FL setups.

7. **The work is well motivated** (Acknowledged by Reviewers **nuGC, 6dpE**) and the **writing is clear and easy to follow** (Acknowledged by Reviewers **nuGC, JN2W**).


##**B. Discussions on Limitations**
We thank the reviewers for the suggestions. We will include a dedicated section to discuss the limitations of our paper, as outlined below.

*Limitations*:  (i) Our research primarily explores visual recognition tasks within the federated learning domain, and we have not explored potential downstream tasks, such as object detection and segmentation.
It's worth noting that visual recognition models often serve as backbones for extracting input image features in downstream tasks, thereby suggesting that our observed performance trends could likely extend to these downstream tasks. We plan to address this research gap in future studies.

(ii) Furthermore, our approach did not design a completely new model from scratch. Instead, we demonstrated our results utilizing pre-existing models and components. Our study also underscores the limitations of employing established optimizers (e.g., FedAVG, FedAVGM, FedProx, SCAFFOLD) along with modern network architectures. However, we abstain from plunging into the creation of innovative optimization techniques. Instead, we focus on evaluating diverse architectural families on many practical federated learning datasets and optimizers. As a result, our investigation leaves an intriguing avenue for future research, integrating our findings to develop dedicated architecture and optimizers tailored specifically for federated learning (FL) scenarios. By exploring this direction, we anticipate the potential to further enhance the performance and efficiency of FL systems.

---

### Decision · Program_Chairs · 2023-09-21

**Decision:**

Accept (poster)

**Comment:**

Five experts reviewed the paper, and all recommended the paper for (borderline or weak) acceptance. The reviewers appreciated the careful and extensive experiments and found some results interesting. The study can inspire and benefit future work in federated visual recognition. However, the reviewers also raised concerns and/or suggestions for improving the paper, which the authors are encouraged to consider when revising the paper. The decision to recommend the paper for acceptance.